# Modeling Average Pressure and Volume Fraction of a Fluidized Bed Using Data-Driven Smart Proxy

**Amir Ansari [1],\*, Shahab D. Mohaghegh [1], Mehrdad Shahnam [2] and Jean-François. Dietiker [3]**

[1]  Petroleum & Natural Gas Engineering Department, West Virginia University; Morgantown,
    WV 26506-6070, USA
[2]  National Energy Technology Laboratory, Department of Energy, Morgantown, WV 26505, USA
[3]  West Virginia Research Corporation, Morgantown, WV 26506, USA
\*  Correspondence: amansari@mix.wvu.edu; +1-(304)-276-7437

**Abstract:** Simulations can reduce the time and cost to develop and deploy advanced technologies and enable their rapid scale-up for fossil fuel-based energy systems. However, to ensure their usefulness in practice, the credibility of the simulations needs to be established with uncertainty quantification (UQ) methods. The National Energy Technology Laboratory (NETL) has been applying non-intrusive UQ methodologies to categorize and quantify uncertainties in computational fluid dynamics (CFD) simulations of gas-solid multiphase flows. To reduce the computational cost associated with gas-solid flow simulations required for UQ analysis, techniques commonly used in the area of artificial intelligence (AI) and data mining are used to construct smart proxy models, which can reduce the computational cost of conducting large numbers of multiphase CFD simulations. The feasibility of using AI and machine learning to construct a smart proxy for a gas-solid multiphase flow has been investigated by looking at the flow and particle behavior in a non-reacting rectangular fluidized bed. The NETL's in house multiphase solver, Multiphase Flow with Interphase eXchanges (MFiX), was used to generate simulation data for the rectangular fluidized bed. The artificial neural network (ANN) was used to construct a CFD smart proxy, which is able to reproduce the CFD results with reasonable error (about 10%). Several blind cases were used to validate this technology. The results show a good agreement with CFD runs while the approach is less computationally expensive. The developed model can be used to generate the time averaged results of any given fluidized bed with the same geometry with different inlet velocity in couple of minutes.

**Keywords:** fluidized bed; computational fluid dynamic; uncertainty quantification; machine learning; artificial neural network; data-driven smart proxy

## 1. Introduction

Fossil fuel continues to be a reliable source of energy for power generation in the United States and worldwide. Technologies such as chemical looping and gasification aim to reduce the carbon emissions of fossil fuel-based power plants. Simulations can reduce the time and cost to develop and deploy such advanced technologies and enable their rapid scale-up. Simulations can be used to test new designs to ensure reliable operation under a variety of operating conditions. However, to ensure their usefulness in practice, the credibility of the simulations needs to be established with uncertainty quantification (UQ) methods. To this end, the National Energy Technology Laboratory (NETL) has been applying non-intrusive UQ methodologies to categorize and quantify uncertainties in computational fluid dynamics (CFD) simulations of gas-solid multiphase flows, which are encountered in fossil fuel-based energy systems [1–4]. Gas-solid flows are inherently highly unsteady

and chaotic flows within which sharp discontinuity can exist at the interface between the phases. The challenge in CFD simulation of gas-solid flows is to adequately resolve the structures that exist at different spatial and temporal scales in an inherently transient flow. Additionally, in reacting gas-solid flow simulations, small time-steps are needed to not only resolve the temporal scales of the flow, but also to ensure the numerical stability of the solution. A rule of thumb for adequate spatial resolution is for the grid spacing to be about 10 times the particle diameter [5]. The grid requirement for maintaining such a ratio of grid size to particle diameter for smaller size particles makes such simulations computationally costly and impractical [4]. Recent work at the NETL [4] has shown the number of simulations required for uncertainty quantification can easily exceed dozens of simulations. The spatial and temporal resolution requirements for multiphase flows make CFD simulations computationally expensive and potentially beyond the reach of many design analysts.

It is clear that a paradigm shift in simulation technology is needed in order to make reacting gas-solid flow CFD simulations with appropriate grid resolution more practical for design and optimization purposes during design scale up. To accelerate the design and analysis process, high-fidelity surrogate models that can capture the flow behavior of the design under consideration can be utilized. Surrogate models are increasingly being used in design exploration, optimization, and sensitivity analysis. Advances in big data analytics and machine learning enable creation of data-driven meta-models, which can faithfully duplicate the behavior of the data that were used for their construction. This new technology has been successfully applied in the upstream petroleum industry [6–9]. Smart proxy modeling takes advantage of pattern recognition capabilities of artificial intelligence and machine learning to build powerful tools to predict the behavior of a system with far less computational cost compared to traditional CFD simulators.

The goal of this research project is to build a smart proxy model constructed from simulation data generated by high-fidelity CFD models to, in effect, replace the use of computationally expensive CFD for the design space under study for further analysis, optimization, and uncertainty quantification. The goal of the portion of research outlined in this article is to establish proof-of-concept for the application of this technology to computational fluid dynamics (at the cell level) and show the capabilities of smart proxy in an up-scaled model (layer level). A smart proxy model, which is constructed from simulation data generated by high fidelity CFD models, can in effect replace the use of computationally expensive CFD for the design space under study and further analysis and optimization. The smart proxy can be used to perform uncertainty quantification analysis to quantify errors and uncertainties that are inherent in any simulation and to quantify uncertainties in the output variables that result from the uncertainties in the input variables. The smart proxy could potentially enable the user to explore the performance of the design well beyond the CFD simulation time window. In other words, a few hundred seconds of CFD simulation time can be used to construct a smart proxy, which can be used to explore the design performance of the unit after many hours of performance. The uniqueness of this approach is in:

1. Developing a unique engineering-based data preparation technology that optimizes the training of the neural networks. This innovative technique incorporates supervised fuzzy cluster analysis to:
   a. Identify the most influential parameters for the training process, and
   b. Identify the optimum partitioning of the data for training, calibration, and validation.
2. Using an "ensemble-based" approach to building the smart proxy, taking advantage of multiple neural networks and intelligent agents to accomplish the objectives of the project.

The idea of using smart proxy and artificial intelligence (AI) in fluid mechanics goes back decades, especially in weather forecasting [10] and aerospace engineering [11]. Recently, Shahkarami and Mohaghegh [9] used an artificial neural network (ANN) to model the pressure and saturation distribution in a reservoir that was used for $CO_2$ sequestration. This problem required a large number of time steps for simulating $CO_2$ injection and storage using commercial software. They ran 10 different cases in numerical reservoir simulator and used the results as the input for ANN. They have shown that ANN can be used as a powerful tool for multiphase flow simulation in the oil and gas industry.

Esmaili et al. [12] incorporated a newly developed AI-based reservoir modeling technology known as Data-Driven Reservoir Modeling [13] to model fluid flow in shale reservoirs using detailed well logs, completion, and production data. Understanding the behavior of the shale reservoir could make conducting the hydraulic fracture much easier. Moreover, this method is able to perform history matching on the production data. Kalantari-Dehghani et al. [14] coupled numerical reservoir simulation with AI methods to develop a shale proxy model that was able to regenerate numerical simulation results in just a few seconds. They introduced three different well-based tier systems to achieve a comprehensive input data-set for the ANN.

Sun and Durlofski [15] used a data-spaced inversion process to predict future reservoir performance using historical data. They utilized principal component analysis and pattern mapping to better align the data distribution with Gaussian distribution. Satija et al. [16] used a combination of statistical and machine learning techniques to predict the reservoir performance without performing history matching to lower the computational cost and uncertainty associated with the history matching process. They used functional data analysis, principal component analysis, and canonical correlation analysis to find the linear pattern between all of the variables in the history and the future of the reservoir. Jeong et al. [17] established a new method by employing artificial neural network and support vector regression to find the relationship between reservoir variables and production prediction. They also used this technique for uncertainty quantification of future reservoir performance.

Ansari et al. [18–20] established the viability of using machine learning and neural network to construct a smart proxy for a fluidized bed based on CFD data from the same fluidized bed. More details on the approach and the steps taken can be found in a thesis by Amir Ansari [21]. Hosseini Boosari [22,23] used a similar approach to model the behavior of dam break flow with the goal of reducing the computational time for the fluid flow simulations by developing a smart proxy model.

## 2. Materials and Methods

A smart proxy is a kind of data-driven model that depends on the availability of the data. Numerical simulation results were used as the ground truth to train this smart proxy. The NETL in-house multiphase solver, Multiphase Flow with Interphase eXchanges (MFiX), has been used to generate simulation data for the rectangular fluidized bed.

### 2.1. MFIX

Multiphase flows are a component of many processes in power generating and chemical processing industries. As expressed earlier, CFD is a valuable tool for designing and optimizing the processes and reactors used in these industries. The NETL has been at the forefront of developing CFD modeling tools that can help engineers and designers improve the performance of processes such as gasification and chemical looping. The MFiX suite of CFD software [24] is open-source, general purpose multiphase CFD software suitable for modeling the hydrodynamics—along with heat transfer and chemical reactions—for a wide spectrum of flow conditions (dilute to dense).

In the present work, the MFiX-TFM (MFiX- Two Fluid Model) is used to model a rectangular 3D fluidized bed. MFIX-TFM, which is based on kinetic theory of granular flow, models both the gas phase and particulate phase as interpenetrating continuous phases. The governing equations employed for the conservation of mass and momentum for each phase ($m$ = g for gas phase and $m$ = s for solid phase) are:

$$\frac{\partial}{\partial t}(\varepsilon_m \rho_m) + \nabla.(\varepsilon_m \rho_m \vec{v}_m) = \sum_{\substack{n=1 \\ n \neq m}}^{N_m} R_{mn} \tag{1}$$

$$\frac{\partial}{\partial t}(\varepsilon_m \rho_m \vec{v}_m) + \nabla.(\varepsilon_m \rho_m \vec{v}_m \vec{v}_m) = \nabla.(\bar{\bar{S}}_m) + \varepsilon_m \rho_m \vec{g} - \sum_{\substack{n=1 \\ n \neq m}}^{M} I_{mn} \tag{2}$$

Where

$\varepsilon_m$  is the phase volume fraction

$\rho_m$  is the phase density

$\vec{v}_m$  is the phase velocity vector

$R_{mn}$  is mass transfer between phases

$\bar{\bar{S}}_m$  is the phase stress tensor

$I_{mn}$  is the interaction force representing the momentum transfer between the phases

The closure terms for the solid phases are obtained through kinetic theory of granular flow. Detailed information on the constitutive relationships used to model momentum exchange between the phases along with the solid stress model incorporated in MFiX-TFM can be obtained from MFiX online documents [25,26] and Appendix A.

Equations (1) and (2), which form a system of nonlinear partial differential equations (PDEs) are discretized temporally and spatially using implicit Euler-based scheme (first order) and second order smart discretization scheme respectively. An iterative algorithm is used in MFiX to solve this system of PDEs. Figure 1 illustrates the solution sequences used in MFiX for solving Equations (1) and (2). To construct the smart proxy, it is important to know this sequence.

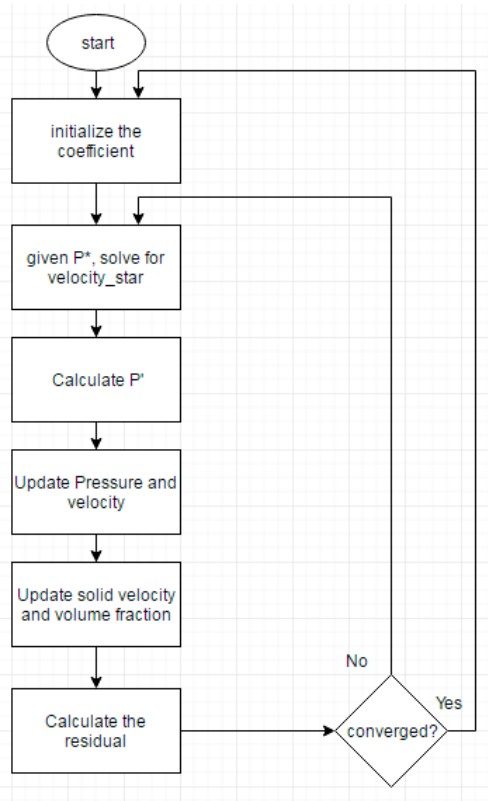

**Figure 1.** Multiphase Flow with Interphase eXchanges (MFiX) solution algorithm

## 2.2. CFD Simulation Setup

A schematic of the rectangular fluidized bed used in this study is shown in Figure 2. The fluidized bed, which is 0.12 (m) by 0.72 (m) by 0.12 (m) in the X, Y and Z directions has an initial bed height of 0.12 m and an initial bed voidage of 0.42. The bed material has a density of 2000 kg/m³ and a diameter of 400 μm. Inlet air velocity is set to 0.6 m/s and is uniformly distributed across the bottom inlet. Uniform structured hexahedral mesh has been used for discretization. The spatial grid resolution is 4.4 mm (11 particle diameters) in all directions and is based on a grid resolution study that was carried out for four different grid levels, shown in Table 1. This is in line with Fullmer and

Hrenya's [5] work showing a grid spacing as small as 10 particle diameters is needed for numerical accuracy.

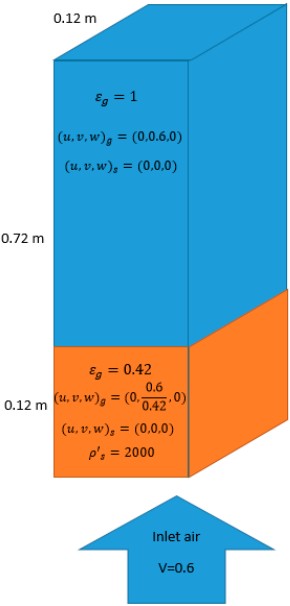

**Figure 2.** Geometry and initial condition of the problem.

**Table 1.** Different grid size and the number of cells.

| Grid Classification | Grid No. (X × Y × Z) | Grid Resolution | No. of Cells |
|---|---|---|---|
| Coarse | 8 × 48 × 8 | 15 mm | 3,072 |
| Medium | 12 × 72 × 12 | 10 mm | 10,368 |
| Fine | 18 × 108 × 18 | 6.6 mm | 34,992 |
| Very Fine | 27 × 162 × 27 | 4.4 mm | 118,098 |

*2.3. Data Preparation*

MFiX outputs all relevant information such as gas and solid velocities, voidage, and pressure field for the entire domain (Table 2 shows all 9 output parameters). The output data from MFiX is used as the input and output data for the training, calibration, and validation of the ANN. The exact location of each grid is important for the smart proxy to establish the spatial correlation (will be discussed in Section 2.3.1). In MFiX, each control volume is represented by its X, Y and Z location (I, J, and K indices). An additional numerical index is defined in MFiX that is unique to each control volume. ParaView, which is open-source visualization software, is used to extract the required data from MFiX and convert it to multiple text files. Data is then reorganized to serve as the input for the ANN, which will be discussed in detail in later sections. Each column represents one output parameter and each row corresponds to one cell. (Every time-step has one text file that contains nine columns and 118,098 rows.) The input to the ANN is all the data at time-step t while the output will be one or more parameters at next time-step ($t + 1$). In this approach, the network will learn what the output should be, given a set of input data. When the learning process has been completed, the deployment process (prediction) will be performed.

**Table 2.** Multiphase Flow with Interphase eXchanges (MFiX) output variables used in artificial neural network (ANN) training.

| Symbol | Description |
|--------|-------------|
| $\varepsilon_g$ | Gas volume fraction |
| $P$ | Gas Pressure |
| $P_s$ | Solid Pressure |
| $u_g$ | Velocity of gas in x direction |
| $v_g$ | Velocity of gas in y direction |
| $w_g$ | Velocity of gas in z direction |
| $u_s$ | Velocity of solid in x direction |
| $v_s$ | Velocity of solid in y direction |
| $w_s$ | Velocity of solid in z direction |

### 2.3.1. Tier System

In order to communicate all the required information to the ANN, so that it can have a reasonable understanding of the state of the process and learn in an effective manner, a tier system was developed based on the grids in the CFD simulation (uniform structured hexahedral mesh). Each cell is in contact with 26 surrounding cells; six of them have the surface contact with the original cell (Tier 1), 12 have line contact with the original cell (Tier 2), and the remaining eight have point contact with the original cell (Tier 3).

Like any numerical method, the value of each variable in each cell is correlated to the variable value in the surrounding blocks. With that idea in mind, the ANN will not only learn from the nine parameters (Table 2) in a cell, it will also learn from the surrounding "*Tier*" cells. Figure 3 shows the Tier 1 structure, in which the main (focal) cell is surrounded by its six neighboring cells. For this case, the nine parameters of the original cell and nine parameters of the Tier 1 cells make 63 parameters ((6+1) × 9), which are the input for the ANN. Depending on the complexity of the problem and spatial and temporal correlations between different tiers and the center cell, more or less input parameters might be required.

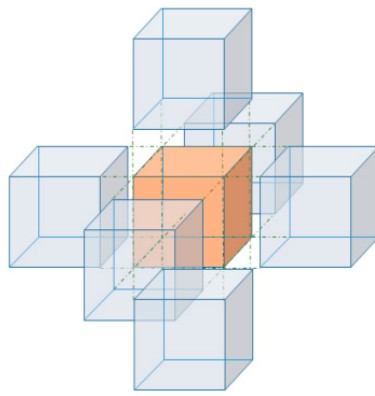

**Figure 3.** The tier system with the six cells in surface contact with the focal cell.

### 2.3.2. Input Matrix

It is not sufficient to consider only the values of each parameter in a focal cell and the related tiers in the input matrix; the location of each cell in the geometry is also crucial for the network to learn the behavior of the process and perform pattern recognition. Adding the location as an input helps the system understand the spatial correlation among different parameters as well. On the other hand, walls (boundary conditions) have an important impact on the flow pattern; therefore, the location of walls with respect to the focal cell should also be included in the inputs in order to train the ANN. To accommodate these ideas, six different distances to the wall confinements (top, bottom,

east, west, north, and south) are considered in order to define the exact location of each focal cell and the parameters associated with each cell. By adding these 6 distances to the previous 63 parameters (nine parameters of seven cells—the focal cell plus six tier one cells), the total number of parameters used as inputs becomes 69, as shown in Figure 4. So, the dimension of input matrix is 69 by 118,098 (i.e., number of parameters multiplied by the number of cells).

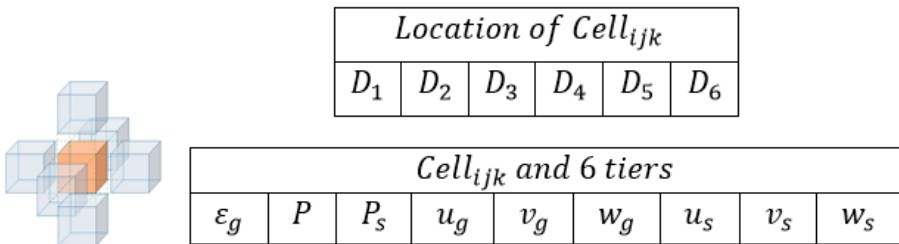

**Figure 4.** Sixty-nine parameters of artificial neural network (ANN).

### 2.3.3. Neural Network Architecture

Each artificial neural network consists of an input layer, one or more hidden layers, and an output layer. The input and output parameters are chosen based on the nature of the problem and the property to be predicted. The previous section described how the number of input parameters were selected to be 69. The output of the ANN is one of the parameters (Table 2) at the next time-step.

There is no clear guideline on how many hidden layers and neurons are required at each layer for a given problem. A rule of thumb indicates that the number of neurons in the first hidden layer should not be less than the number of input parameters. For the first try, one hidden layer with 100 neurons is considered where 69 parameters are used as inputs and only one parameter is selected as output.

The initial network characteristics are shown in Table 3 (these characteristics will be optimized). Feed-forward back propagation algorithm is used for the training. In this algorithm, input data go to ANN (feed-forward), and the output values are calculated based on the initial ANN's weights. The error is obtained by comparing the actual target and the values predicted by the ANN. The error is then sent back to the ANN (back propagation) to update the initial weights; this is called one epoch (one feed-forward and one back propagation). The adaption learning function is LEARNGDM, which is the regular gradient descent method that will be used to find the optimum ANN's weights. Gradient descent is an iterative optimization technique that starts with an initial guess, then the gradient (slope) at the initial guess is calculated to identify the direction to local minima. Performance function is the mean square error (MSE) (Equation 3). The performance function is used to evaluate the quality of the trained model and to stop the training if no improvement takes place. The transfer function for the hidden layer and the output layer was chosen to be tangent sigmoid function (TANSIG), shown in Figure 5a. Figure 5b shows log sigmoid (LOGSIG) transfer function, which is widely used. The transfer function (or activation function) gives the artificial neural network the ability to capture the non-linearity in the output; an ANN without a transfer function is just a multiple linear regression model. These hyper-parameters should be selected following an extensive optimization process.

**Table 3.** Neural Network characteristics.

| Network Type | Feed-forward Back propagation |
|---|---|
| Training Function | Levenberg-Marquardt |
| Adaption Learning Function | LEARNGDM |
| Performance Function | MSE |
| Transfer Function | TANSIG |

$$MSE = \frac{1}{n}\sum_{i=1}^{n}\left(y_{actual} - y_{predicted}\right)^2 \tag{3}$$

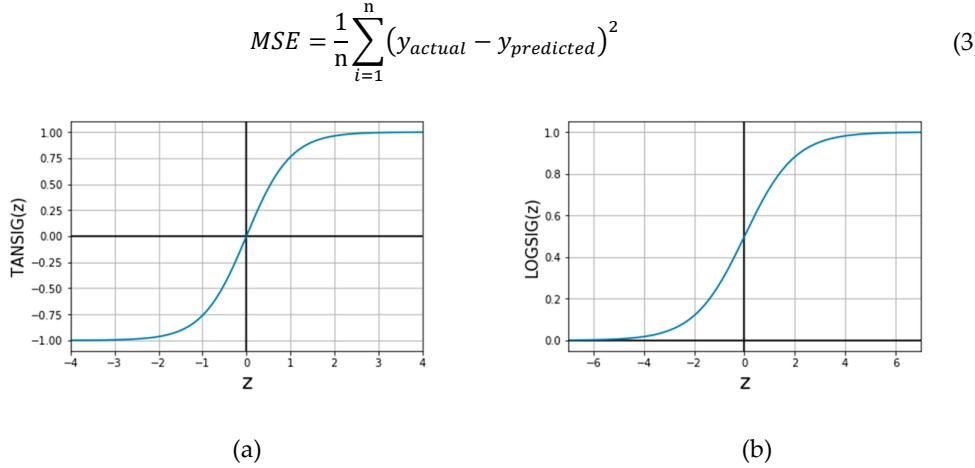

(a)　　　　　　　　　　　　　　　　　　　　(b)

**Figure 5.** Transfer functions: (**a**) tangent sigmoid function (TANSIG), (**b**) log sigmoid (LOGSIG).

2.3.4. Data Partitioning

A good AAN is a model that learns the pattern in the given data-set while it is able to predict the behavior of a new given data-set; this model is called "Just Right." If the ANN does not learn the pattern in the data very well, the model is called "Under-fit." If the ANN learns the pattern of the data very well and with a very small error, but it is not able to predict the behavior of a new given data-set, the model is called "Over-fit." Under-fitting occurs for various reasons, one of which is lack of information (the model should have more parameters and more examples). Overfitting occurs when the network learns to replicate almost all of the data points, but when it comes to prediction, the model performs poorly for new given data. In other words, the model has memorized all the data points rather than learning to predict the behavior of a new data-set.

To overcome the over-fitting problem, only a portion of the data is used to train the network, and the rest of the data are kept outside of the training as a criteria to stop the training process when the model is "Just Right" and validate it after the training is complete. The remaining data points, which the model has not seen in the training process, are further divided into two sub groups: calibration and validation.

Training is an iterative process where in each iteration the optimization algorithm tries to advance toward the lower error value. A calibration data-set is used while the training is being carried out. The errors in both the training and calibration data-sets usually decrease at the beginning of the training process; however, sometime during the training process, the error in the calibration data-set stops decreasing while the error in the training data-set continues to decrease. The model at this point is usually the best model because it has provided the lowest possible error for the calibration data-set (blind data-set) while still having an acceptable error for the training data-set.

The validation data-set is used upon the completion of the training process when the best ANN is achieved. Having an ANN model with a low calibration error does not necessarily mean that the ANN is a good predictor (because the best model is already picked when the calibration error is minimum) unless the ANN error in the validation data-set is also acceptable.

The percentage of the data partitioning used for the preliminary study of this project is shown in Table 4. It is important to mention that this partitioning is the preliminary one, and a deeper study will be conducted on the percentage of the data as will be described in the upcoming sections of this report.

**Table 4.** Original data partitioning.

| Data | Training | Calibration | Validation |
|---|---|---|---|
| Percentage of data (%) | 70 | 15 | 15 |

The training process stops based on user defined criteria. The termination criterion could be the total number of iterations, the total time of training, or the number of calibration failures or, as is the case in this work, the termination criteria is a combination of all of the above. The learning algorithm is such that the network learns more after each iteration, so to prevent overfitting or memorization, the calibration error is always checked. If the calibration error increases for a predefined number of iterations, the training stops. Most of the time, calibration is the criterion that makes the training process stop.

## 3. Proof of Concept

MFiX simulation results of the rectangular fluidized bed are used as the training, calibration, and validation data to build a spatio-temporal database for the construction of a fluidized bed smart proxy (Figure 6).

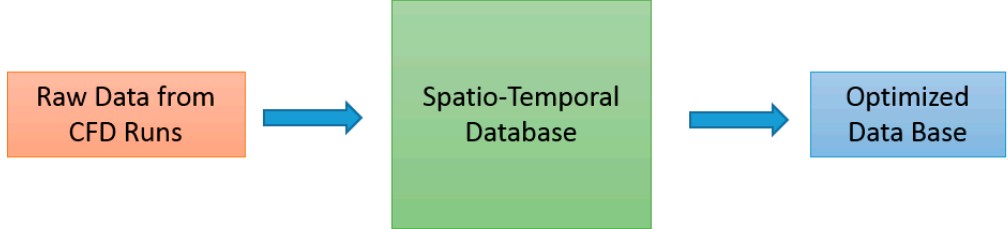

**Figure 6.** Spatio-temporal database and optimized database.

The Spatio-temporal database was created based on the data from simulation runs. This database includes the location and the properties of each cell and the properties of tier cells (69 parameters total) for every time step. Different scenarios are considered in order to reach the final goal of this part of the project, which is to illustrate the feasibility of constructing a smart proxy for a fluidized bed based on data generated from CFD. The term "different scenarios" refers to having different input and output structures and also using different time-steps for the training of ANN, while the training algorithm is kept the same. To accomplish the objective of each scenario, a different subset of the spatio-temporal data is selected, which is the optimized dataset for that specific scenario.

Each scenario has two parts: training process and deployment process. As mentioned earlier, at a given time step, 69 parameters (static parameters and dynamic parameters of time-step t) are used as the input to ANN and one parameter (for instance, gas pressure of time-step $t + 1$) as the output. It is important to emphasize that in the training stage, both input and output are directly from reference data (CFD output results). Figure 7 depicts the training process where the direction of both arrows is toward the ANN, meaning that both input and output are fed to the ANN in the training process. The training process is an iterative method in which the ANN's parameters will be updated in each iteration (epoch).

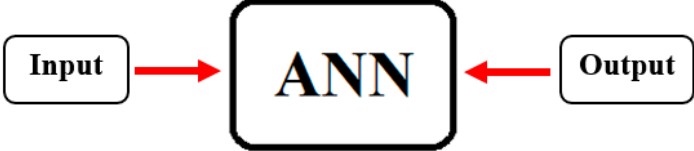

**Figure 7.** Training stage flow chart.

The trained network is then ready for the deployment stage, where data at a given time step is used as the input, and the trained network provides prediction at the next time step (Figure 8). The input of the ANN for each deployment could come from the CFD directly or from the ANN itself. Cascading and non-cascading deployment are defined based on the type of input used for the network, and it will be discussed in detail in the following sections.

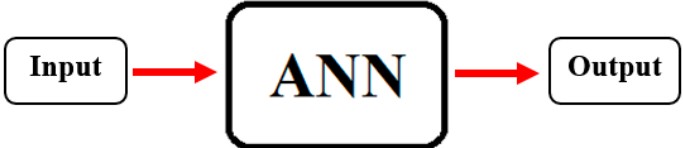

**Figure 8.** Deployment stage flow chart.

As stated earlier, this phase of the current research aims to show the feasibility of constructing a smart proxy based on CFD results for a fluidized bed. As such, the scenarios outlined below show the systematic steps that have been taken from the least complex scenario to the more complex scenarios.

### 3.1. Early Time versus Late Time

The fluidization process, as shown in Figure 9, starts with the bed material moving upward like a slug flow (Figure 9a) until the maximum bed expansion is reached (Figure 9b) and the bed starts to collapse. In Figure 9, the color red indicates high voidage (low solid volume fraction) and the color blue indicates low voidage (high solid volume fraction). The solid flow is symmetrical until the bed collapses, smaller bubbles are formed, and the bed behaves more chaotically (Figure 9c). Ultimately, the bed becomes fully fluidized and chaotic (Figure 9d). In order to see more details within the fluidized bed, the results will be shown in five cross-sectional planes perpendicular to Z axis as shown in the Figure 10.

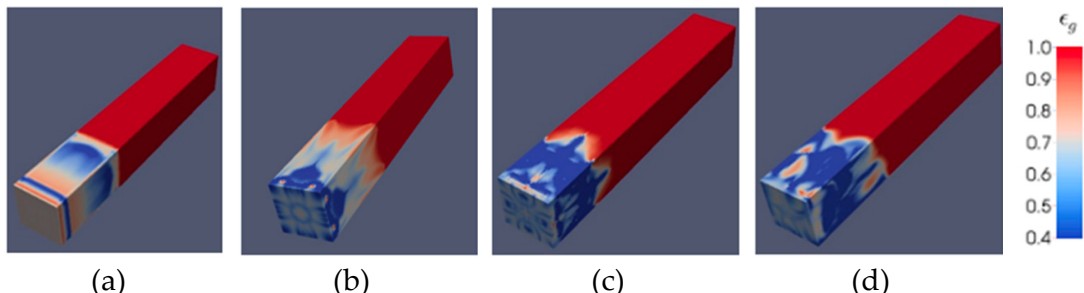

**Figure 9.** Gas volume fraction contours at different times encountered in the fluidized bed (for illustration purposes, the bed is titlted horizontally) (**a**) slug flow, (**b**)maximum bed expansion, (**c**) bed collapse, and (**d**) fully fluidized bed

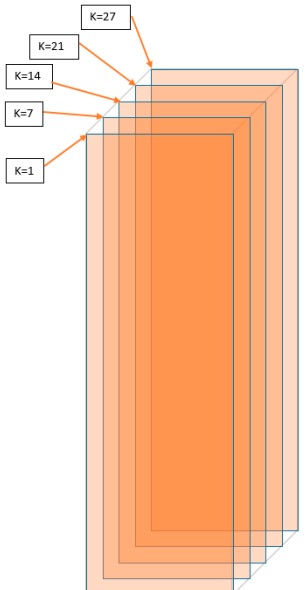

**Figure 10.** Cross-sectional planes, 3 cm apart, where results are presented.

Figure 11a shows time-step 100 when the flow behaves like a slug flow, before instabilities set in and fluidization begins at a later time. As time goes by, changes in flow regime occur and flow becomes more chaotic and heterogeneous, as shown in Figure 11b. It is therefore necessary to investigate how well an ANN can be trained when the degree of heterogeneity in the flow increases. ANN can be trained based on flow encountered at the early stage of fluidized bed operation or at a later time. In both cases, the 69 inputs come from time-step (*t*) and the CFD output is from time-step (*t* + *Δt)* in Figure 12 (*Δt* could be 1, 2 or more). The ANN output could be one parameter from Table 2. Each time-step used for training represents 1 millisecond of simulation time. The purpose of this analysis was to show that the ANN is capable of capturing all the physics involved in different time-steps (different flow regimes).

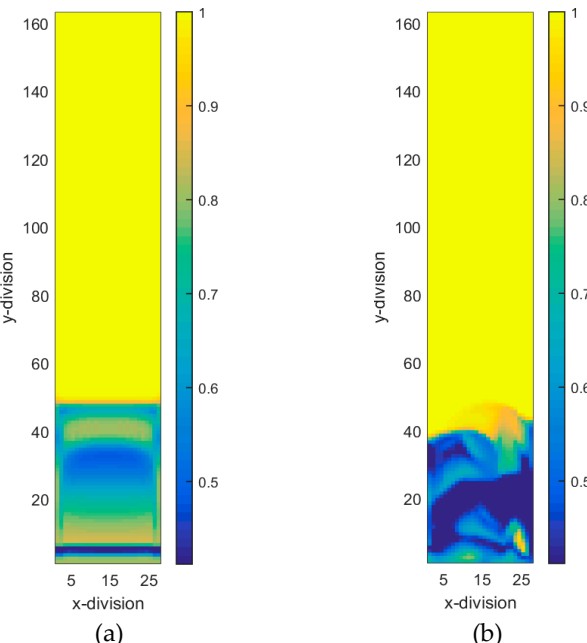

**Figure 11.** Gas volume fraction distribution initially in the bed at time-step = 100 (**a**) and later when it is fully fluidized at time-step = 4000 (**b**).

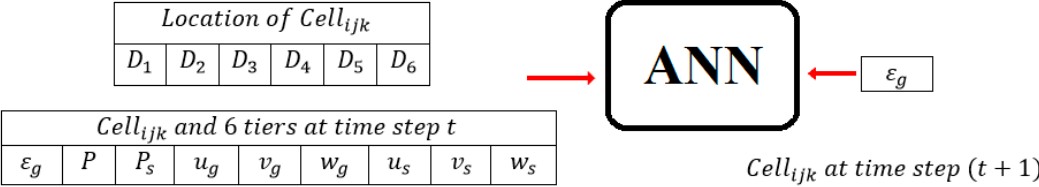

**Figure 12.** Input and computational fluid dynamics (CFD) output parameters used for training.

Time-steps 100 and 101 ($\Delta t$ = 1) were used to train the ANN for gas volume fraction. After the training was completed, to deploy the model, all the time-steps from 101 all the way to 120 were used as input to the ANN and acceptable results were obtained. Figure 13 shows the results of gas volume fraction for time-step 102. The left graph is CFD results, middle graph is smart proxy results, and right graph is the error in percentage. Error is calculated using the following equation.

$$Error = \frac{(CFD_{value} - CFD_{value\,mean})}{(CFD_{value\,max} - CFD_{value\,min})} - \frac{(Smart\,Proxy_{value} - CFD_{value\,mean})}{(CFD_{value\,max} - CFD_{value\,min})} \tag{4}$$

Also, the time steps 4000 and 4002 ($\Delta t$ = 2) were selected to train the network. Similar to the previous scenario, the ANN had only one output—gas volume fraction. For the deployment, all the time-steps from 4002 to 4040 were input to the ANN and acceptable results were obtained. Figure 14 shows the results of smart proxy with maximum error of around 4%. Nine separate ANNs were trained for all the parameters in Table 2 and good results were obtained for all of them.

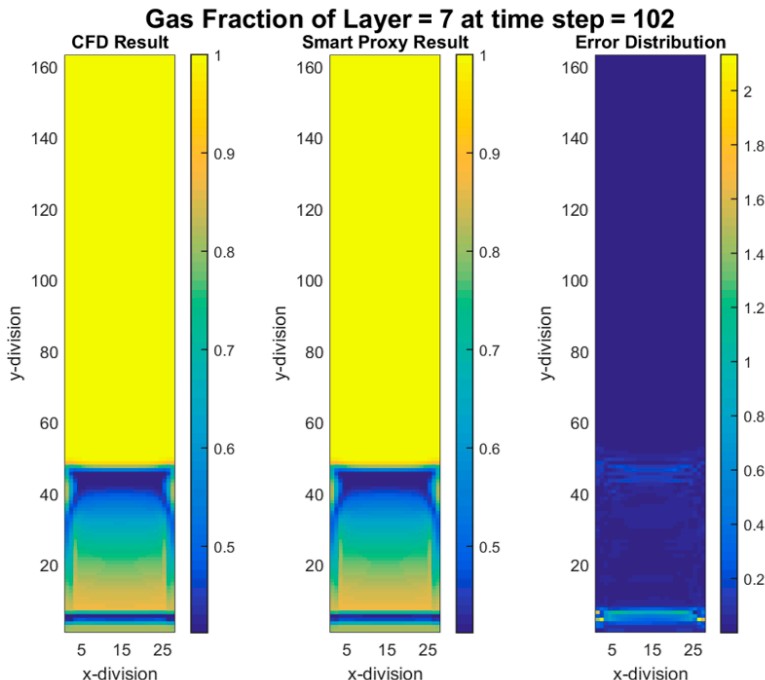

**Figure 13.** CFD and smart proxy results for gas volume fraction at K = 7 cross-sectional plane (time-step 102—early time).

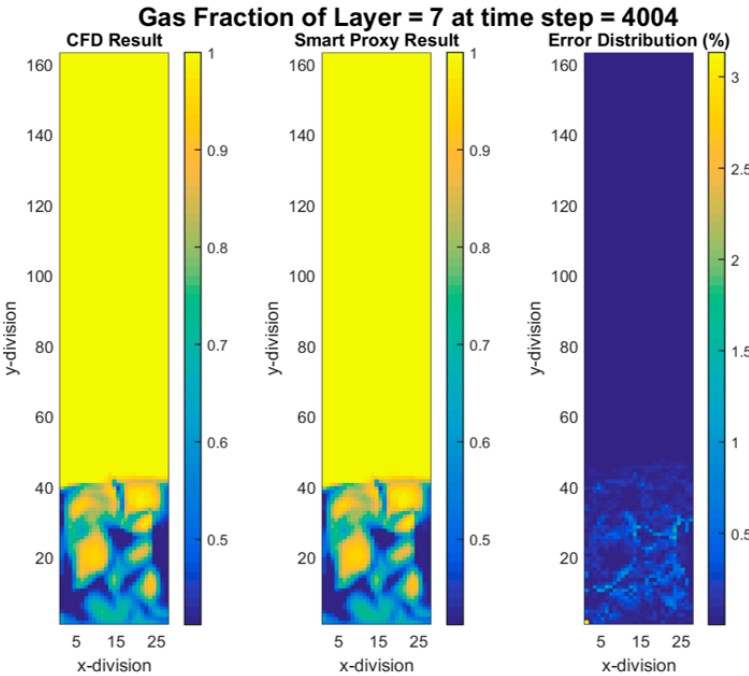

**Figure 14.** CFD and smart proxy results for gas volume fraction at K = 7 cross-sectional plane (time-step 4004—early time).

### 3.2. Cascading versus Non-cascading in Time

Cascading and non-cascading refer to the source of input that is used for the deployment process. If the input comes directly from the CFD simulation model for each deployment stage, then the process is called "non-cascading," shown in Figure 15. In other words, to obtain the result of any desired time-step (t), the previous time-step ($t-1$) should be available from CFD data. If the input of the ANN for each deployment stage comes from the output of previous deployment, then the process is called "cascading," shown in Figure 16. In a cascading approach, the starting time-step should be available from CFD data, the smart proxy predicts the distribution of all of the attributes for the next time-step, and the calculated results are the input to smart proxy for the next time-steps. A detailed discussion on cascading and non-cascading can be found in a technical report by Ansari et al. [18].

The benefit of a non-cascading approach is to check if ANN can perform well if the true input is used. Eventually, the cascading approach is the correct approach to be used for the smart proxy because CFD results are not always available.

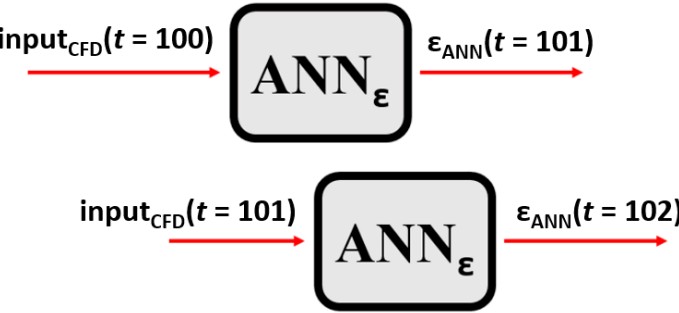

**Figure 15.** The process of non-cascading deployment.

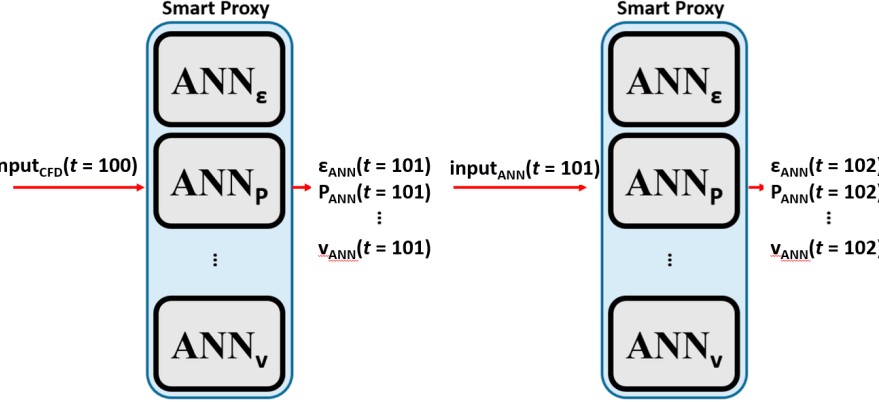

**Figure 16.** The process of cascading deployment.

The same smart proxies were used in the cascading approach for early and late time frames. Figures 17–20 show the contours of gas volume fraction for time steps 4002 (4.002 sec elapsed time) through 4020 (4.02 sec of elapsed time) when flow is fully fluidized. These figures show that percent error increases in time. The error propagation from time-step to time-step can eventually go beyond the user defined tolerance level that forces the cascading deployment process to terminate. In order to overcome the error propagation, more time-steps should be used for training.

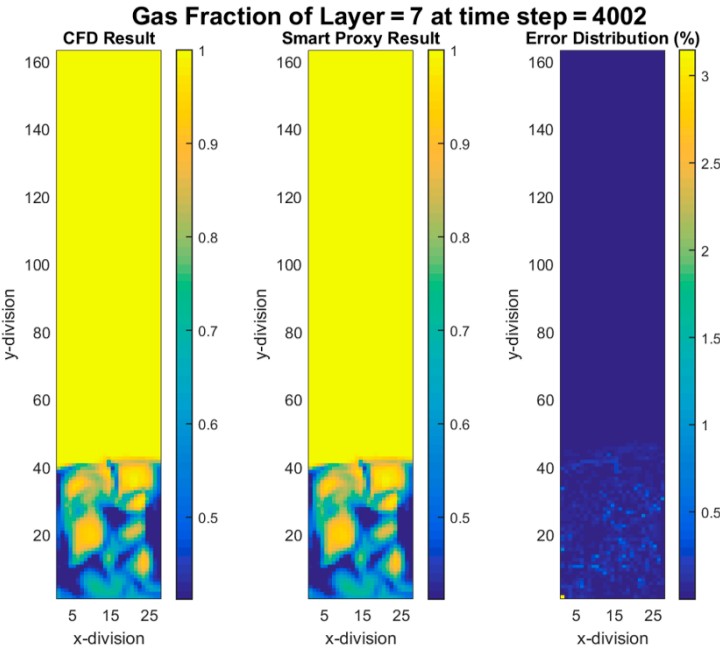

**Figure 17.** CFD and cascading smart proxy results for gas volume fraction at K = 7 cross-sectional plane and time-step = 4002.

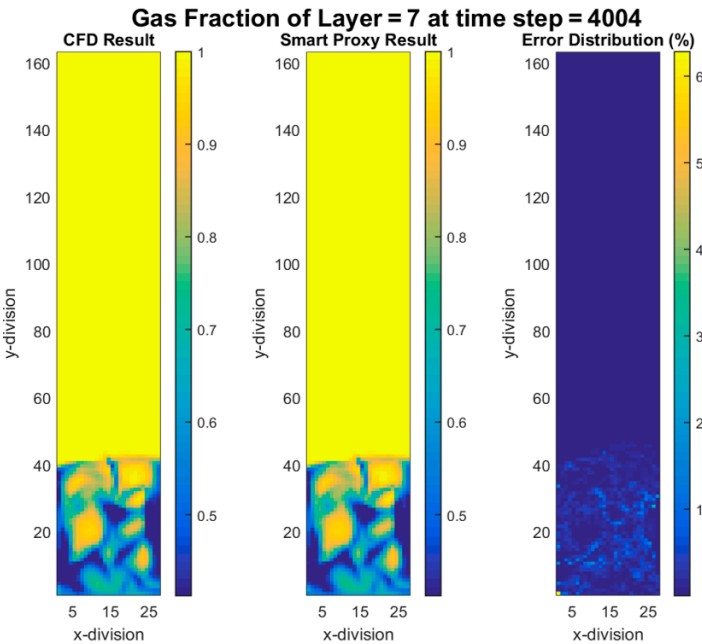

**Figure 18.** CFD and cascading smart proxy results for gas volume fraction at K = 7 cross-sectional plane and time-step = 4004.

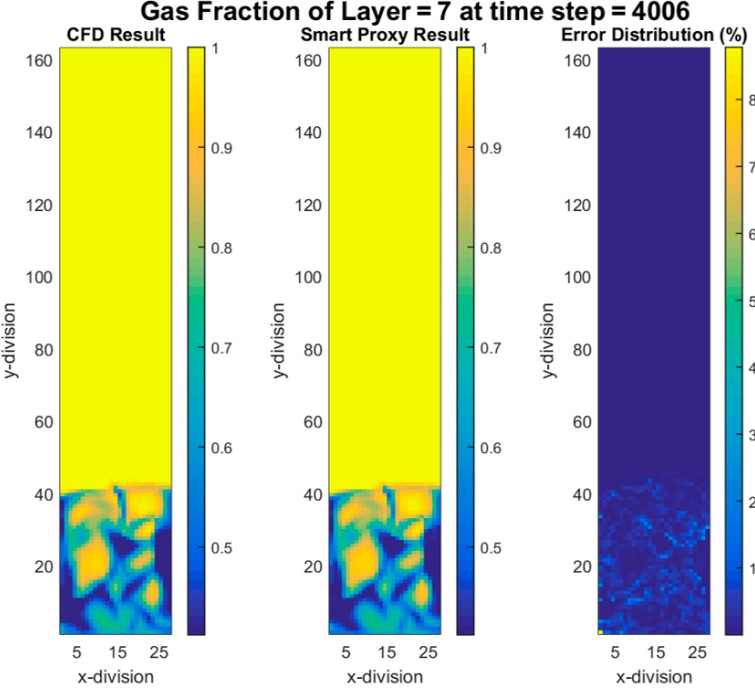

**Figure 19.** CFD and cascading smart proxy results for gas volume fraction at K = 7 cross-sectional plane and time-step = 4006.

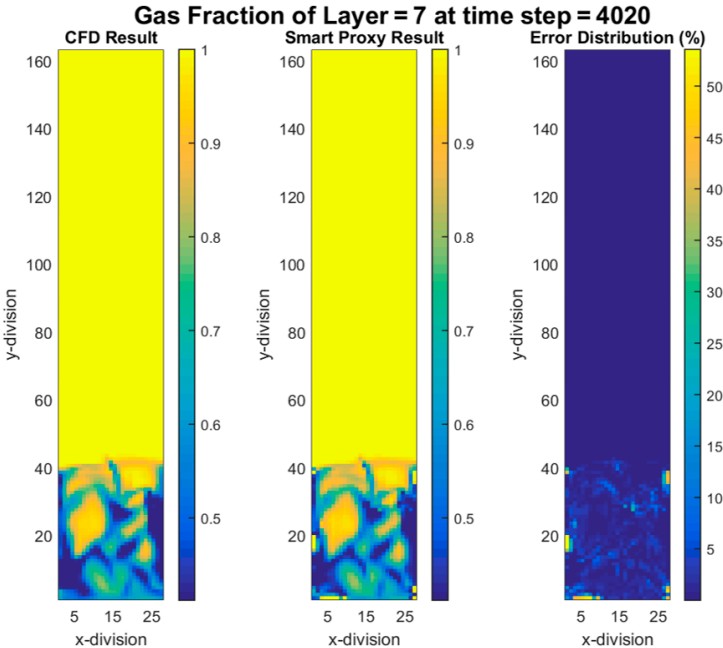

**Figure 20.** CFD and cascading smart proxy results for gas volume fraction at K = 7 cross-sectional plane and time step = 4020.

### 3.3. Training with Multiple Time-steps

The techniques outlined thus far use a single time-step for input and output of the ANN. However, the gas-solid flow undergoes a regime change from a slugging flow at the beginning to a fluidized regime as time goes by. The ANN trained with the data from when the flow field is slugging does not have the predictive capability of capturing the flow dynamics when the bed is fully fluidized. In order to train an ANN, which has a wider range of applicability, the input and output of the ANN must be trained on data from multiple time-steps, capturing many changes taking place in the flow.

Figure 21 shows the input and output pair for the training with three different time-steps: when the flow is slugging at first (time-step of 200), then transitioning (time step 1000) and, finally, fluidizing stage (time step 4000). One more parameter was added to the training data that shows the actual elapsed time for each time-step. Figure 22 shows the gas volume fraction contours in the CFD simulations at time steps 200 (0.2 sec elapsed time), 1000 (1 sec elapsed time) and 4000 (4 sec elapsed time).

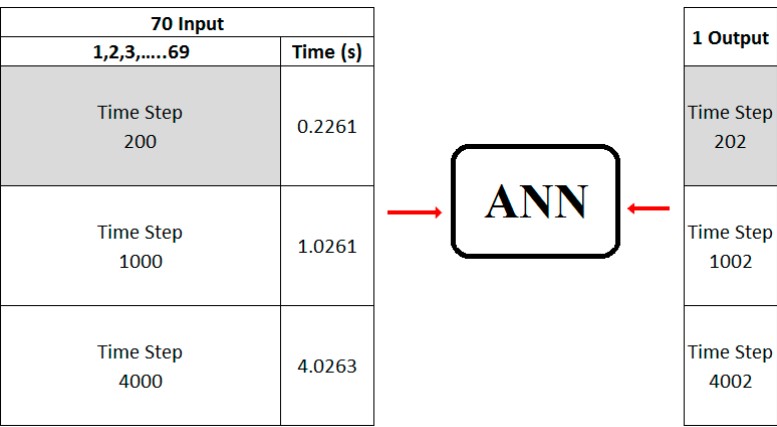

**Figure 21.** Input and output pair for the training.

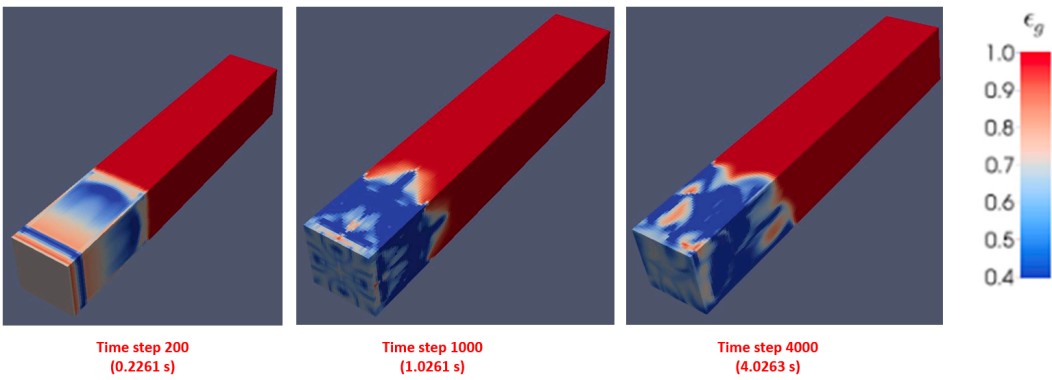

**Figure 22.** Three different time-steps with different flow characteristics.

Time steps 200–202, 1000–1002, and 4000–4002 have been used for the training. The deployment process is then conducted with the trained ANN by inputting time step 200 all the way to time step 4000 (non-cascading approach). The results are presented in Figure 23 in terms of root mean square error (RMSE) of gas volume fraction. Figure 23 shows the RMSE distribution versus time-steps. It is clear that for the time-steps for which there was training data, the amount of error was at a minimum, but in the other time-steps the RMSE had increased. The time steps with larger peaks in Figure 23 point to the need for additional ANN training at those time steps (potentially other dynamic behaviors are taking place in the bed, requiring additional training). The contour plot of gas volume fraction at time step 500, where the RMSE is high (see Figure 23) is shown in Figure 24. It is clear that at around this time step, flow is transitioning from the initial plug flow behavior to a more bubbling flow. This change in the flow regime was not properly captured since the input data to ANN at the training stage did not include any data from the time steps when transitioning was taking place. Figure 25 shows a decrease in RMSE when data from time steps where RMSE peaks in Figure 23 were added to the training data-set.

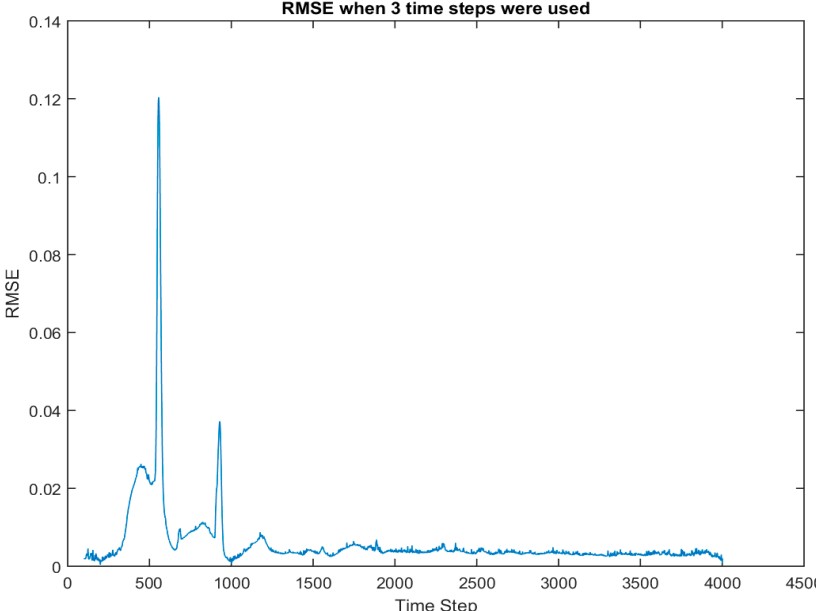

**Figure 23.** Root mean square error (RMSE) distribution over time, when three pairs of data are used for training.

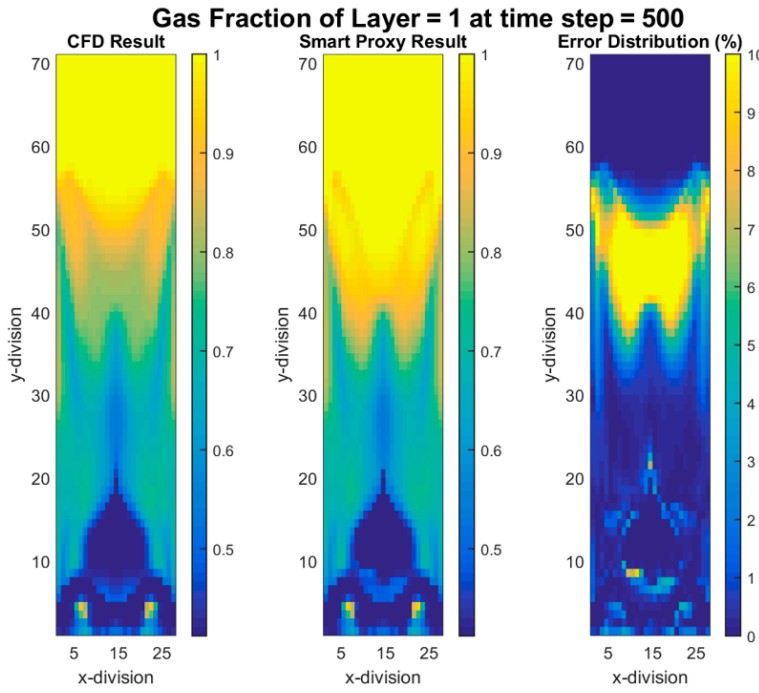

**Figure 24.** CFD and smart proxy results for gas volume fraction at K = 1 cross-sectional plane.

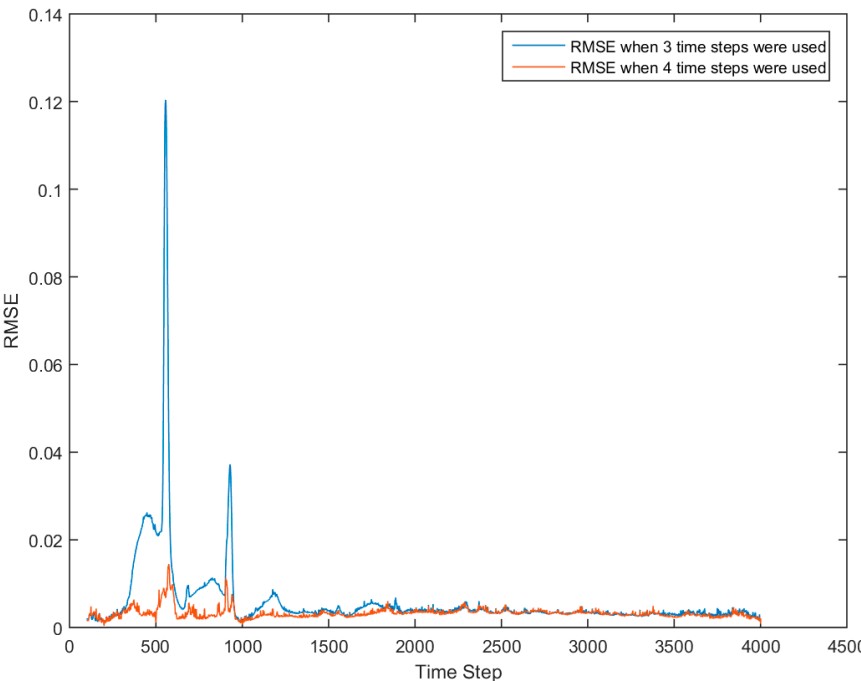

**Figure 25.** RMSE distribution over time, when four pairs of data are used for training.

Although the RMSE is very low in non-cascading approach (Figure 25) for all the times-steps, using the same smart proxy in cascading mode will be accompanied by a large error. This study showed that the smart proxy could be a viable tool for predicting gas-solid flow behavior in a fluidized bed. However, to make the trained ANN more general, the cascading deployment needs further research in order to minimize the error propagation over time. In the next section, the time-step from the training is removed and the smart proxy cascades through location instead of time.

## 4. Model Verification

Several CFD simulation runs with different inlet air velocities for a rectangular fluidized bed are used to create a smart CFD proxy that is capable of replicating the CFD results for a wide variety of inlet velocities. The smart CFD proxy is validated with blind CFD runs (CFD runs that have not played any role during the development (training, calibration, and validation) of the smart CFD proxy). In this section, the ANN is constructed and trained based on cross sectional area average of pressure and volume fraction (Layer Level), which leads to improvements in the training time at the expense of less spatial resolution. The resulting trained ANN provides spatially averaged values of parameters of interest along the length of the fluidized bed. Upon completion of this project, UQ studies that rely on hundreds or thousands of smart CFD proxy runs can be accomplished in minutes.

The inlet air velocity is assumed to be uniform across the fluidized bed inlet (Figure 2) with air discharging into atmospheric pressure at the outlet. The inlet air velocity varies from a minimum value of 0.6 m/s to a maximum value of 1.2 m/s. Several runs with different inlet velocities as depicted in Figure 26 are used to build this smart proxy, the inlet velocities with a green circle are the runs that are being used in the training process.

The goal of this section is to predict the behavior of the fluidized bed at the Layer Level (cross sectional planes along the length of the fluidized bed) at any given inlet air velocity (within the velocity range used for training) at any time. The neural network model will be trained for seven different inlet velocities.

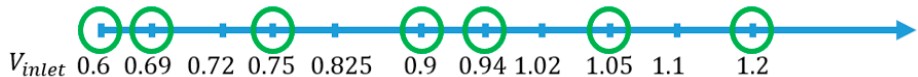

**Figure 26.** All inlet air velocities used in MFIX simulations along with inlet velocities used in ANN training.

The predictive capability of the trained neural network is evaluated using CFD data, which has not been part of the training process (blind test). A blind test is when some of the data that was not used during the training of ANN is used to further validate the predictive capability of the trained ANN (Figure 27). The difference between calibration and validation during the training process and the complete blind test is that the records in the calibration and validation process during the training are chosen randomly from the original dataset; for example, some of the records of inlet velocity of 0.9 are in the training set and some of them are in the calibration and validation set, but all the records of inlet velocity of 0.825 are in the blind test set, which makes the prediction much more difficult.

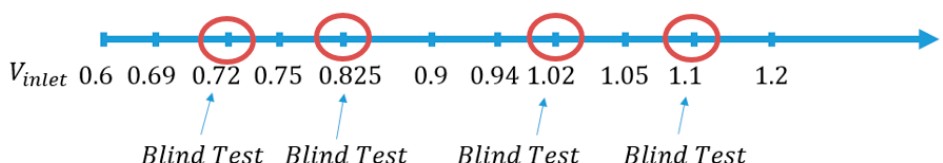

**Figure 27.** All inlet air velocities used in MFIX simulations along with inlet velocities used in ANN blind test.

### 4.1. Layer Level

The training procedure is utilized at the Layer Level. The advantage of training at the Layer Level, in comparison to Cell Level (in the Proof of Concept section), is that ANN needs to predict only one value for each cross section, while in the cell level training, ANN had to predict the values of all the cells in a layer, and obviously it is easier to predict one averaged value rather than 729 (= 27 × 27) values for each cross section. Additionally, by using the layer level approach, the information from upstream and downstream could be used in the training as extra information for the ANN. The average gas pressure at a horizontal plane is used to construct the ANN. Since there are 162 grid points in the vertical direction in the CFD simulations, 162 layers are used for constructing the ANN. Figure 28 shows the difference between the cell based approach, in which an ANN could be

constructed for each computational cell (Figure 28a) and the layer based approach, where an ANN could be constructed for the averaged values across each cross-sectional area (Figure 28b).

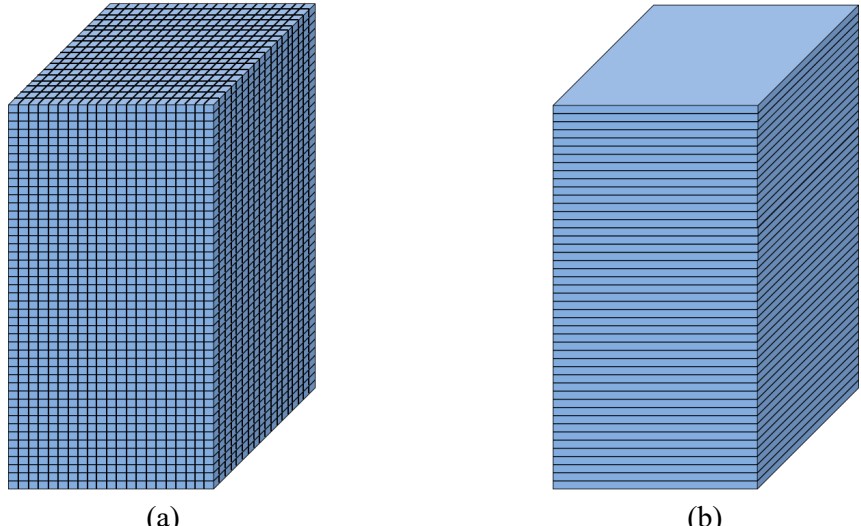

(a)    (b)

**Figure 28.** Different training approaches for ANN at (**a**) Cell Level (**b**) Layer Level.

*4.2. Training for gas pressure using static parameters*

In the Cell Level approach, identification of the cell location was done by using six values (distances to the right, left, top, bottom, north, and south walls, as shown in Figure 29a). In order to define the location of each layer in the Layer Level approach, there are only two distances to the top and the bottom walls (Figure 29b). By using these two distances, we can teach the ANN how close the exit and inlet planes are to each layer, which helps the ANN to handle the boundary condition. In the Layer Level approach, the available static parameters are only four features.

As depicted in Figure 30, a neural network is trained with four static parameters (two distances to the exit and inlet planes to account for boundary effects, the index of the layer, and the inlet velocity) at time step 1400, along with inlet velocity value. Additionally, average gas pressure from CFD results at each inlet velocity is used as output to ANN. For this scenario (Figure 30) at each inlet velocity and time step, there are 162 records, for a total of 1,134 (7 × 162) records for all the inlet velocities.

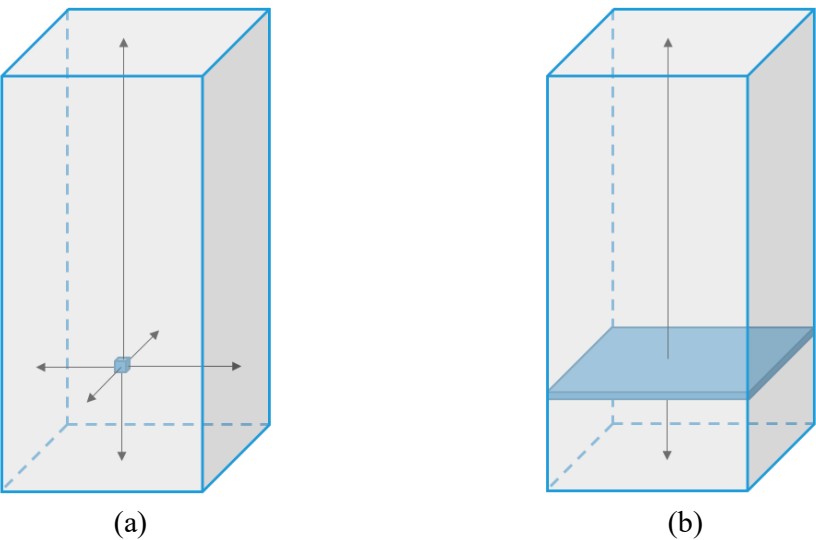

(a)    (b)

**Figure 29.** Distances to the wall in Cell Level (**a**) and in the Layer Level (**b**).

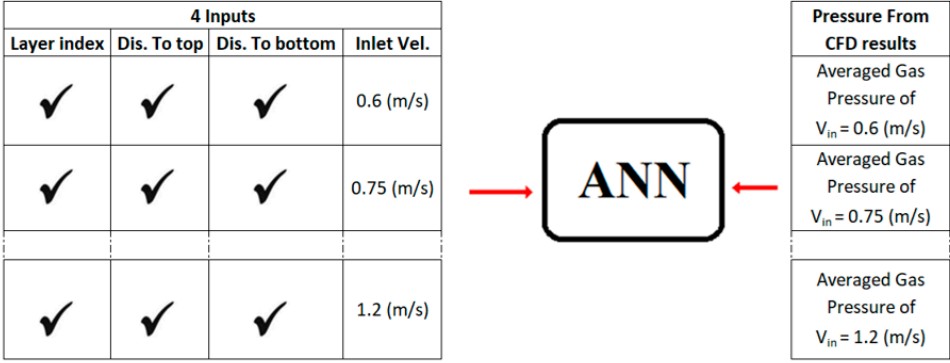

**Figure 30.** Traning for average gas pressure using four static parameters at layer level at each time step.

Some of the ANN hyper-parameters, such as number of hidden layers, number of hidden neurons, transfer function, and the training algorithm have been used in an optimization process to improve the quality of the ANN results. A comprehensive study on these parameters was done by changing all of the hyper-parameters simultaneously and evaluating the performance of the ANN on the blind dataset, and the model with the optimized hyper-parameters was achieved. For instance, the best learning algorithm was the gradient decent method with adaptive learning rate, the best transfer function was LOGSIG, and number of neurons was between eight and 15.

The model has been trained and calibrated. To demonstrate the results, the pressure drop curves are provided for both training cases. Figures 31 and 32 show the comparison between the smart proxy and CFD results for the training data when inlet velocity of $V_{in}$ = 0.6 m/s and 1.2 m/s are examined, respectively. It is clear from these figures that the ANN performs well for the inlet velocities that are part of the training. It is noteworthy to stress that the pressure drop across the bed is a measure of solid inventory (weight of the bed). In the fluidized bed used in this research, the pressure drop across the bed is small (less than 1500 Pa) for all test conditions, since there is only 1.45 kg of solid particles in the bed. The magnitude of pressure drop across the bed has no effect on the ability of the smart proxy to learn from and reproduce the CFD results.

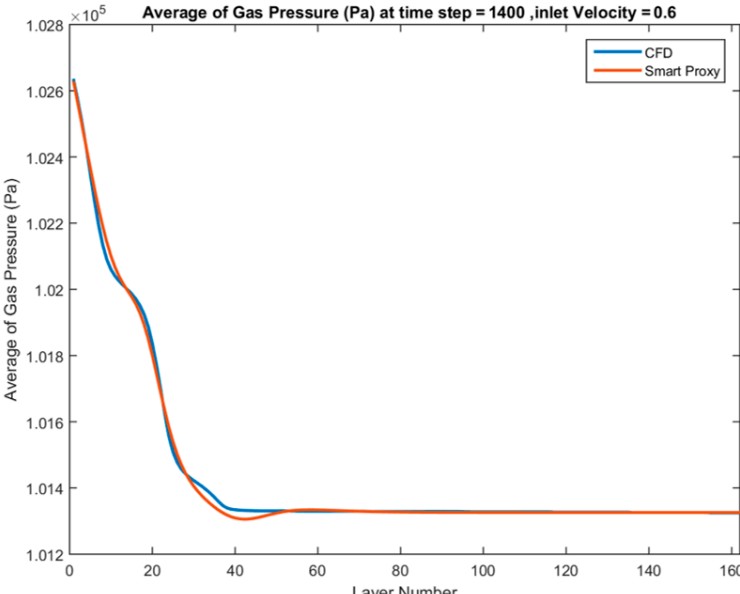

**Figure 31.** Spatially averaged CFD and smart proxy results for gas pressure (Pa) at time step = 1400 and $V_{in}$ = 0.6 m/s (with four static parameters).

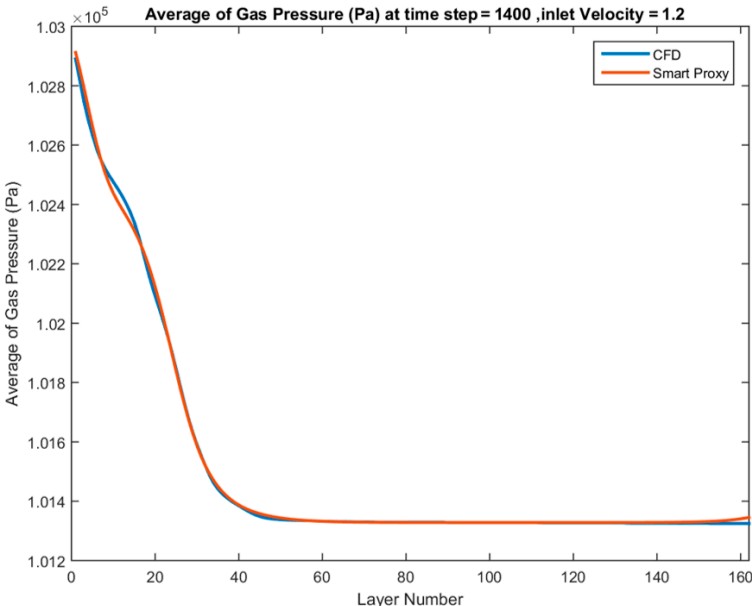

**Figure 32.** Spatially averaged CFD and smart proxy results for gas pressure (Pa) at time step = 1400 and $V_{in}$ = 1.2 m/s (with 4 static parameters).

Once a good training is achieved, ANN is deployed on several blind test cases to evaluate the predictability of the ANN model on the inlet velocities that have never been seen by the model in the training process. Figures 33–35 show the quality of the ANN for blind test cases, when inlet velocities of $V_{in}$ = 0.72, 0.825, and 1.02 m/s are used, respectively. Note that these inlet velocities were kept completely out of the training. The pressure values for the free board are almost constant and equal to the exit pressure; thus, to visualize the results, only the bottom of the bed (up to layer 70) will be shown in the next graphs to see the agreement better.

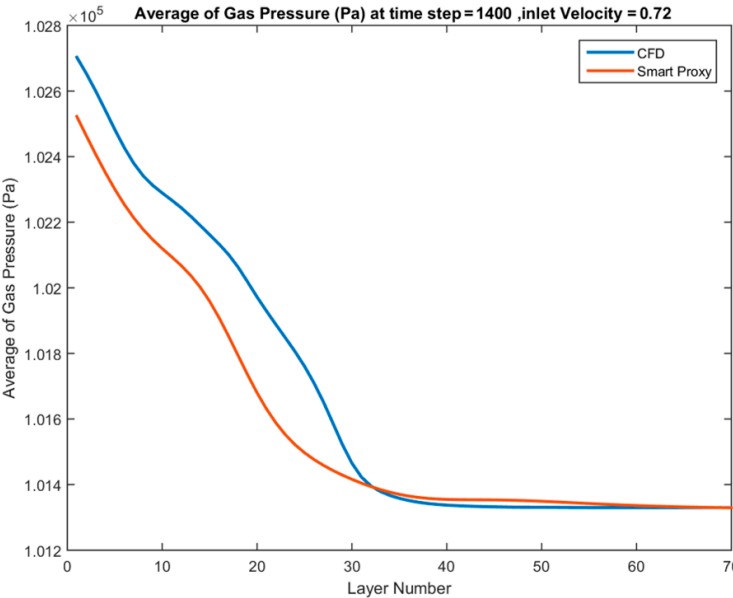

**Figure 33.** Spatially averaged CFD and smart proxy results for gas pressure (Pa) at time step = 1400 for blind test condition of $V_{in}$ = 0.72 m/s (with four static parameters).

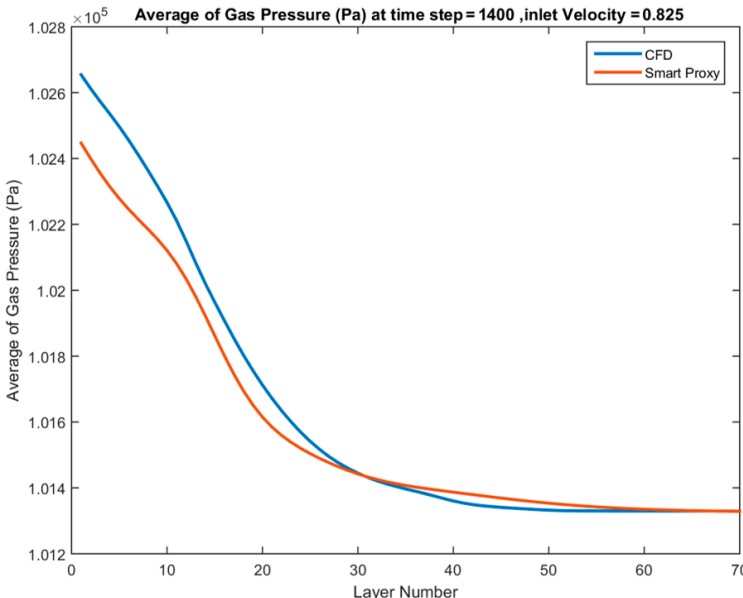

**Figure 34.** Spatially averaged CFD and smart proxy results for gas pressure (Pa) at time step = 1400 for blind test condition of $V_{in}$ = 0.825 m/s (with four static parameters).

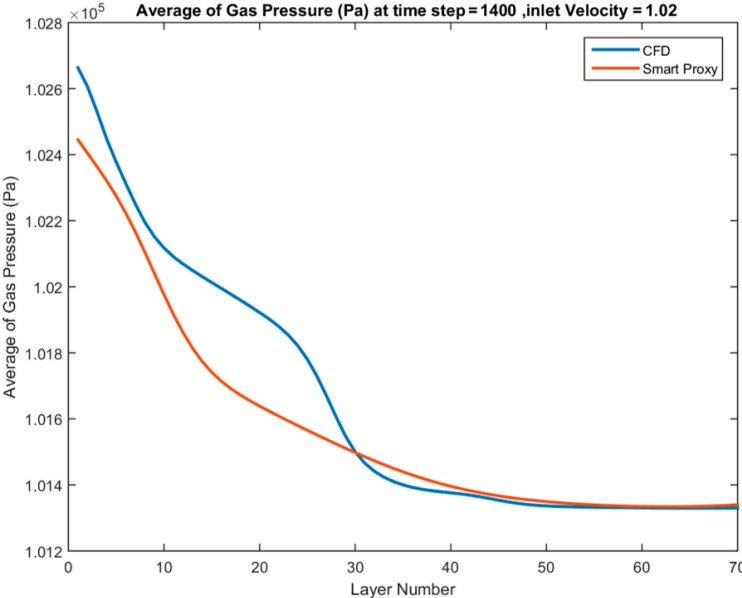

**Figure 35.** Spatially averaged CFD and smart proxy results for gas pressure (Pa) at time step = 1400 for blind test condition of $V_{in}$ = 1.02 m/s (with 4 static parameters).

Although there is good agreement between CFD and smart proxy results in the training cases, there is a poor agreement for the blind test cases. Figures 33–35 show that the ANN is predicting a much more dilute flow in the vicinity of the inlet and the lower portion of the bed compared to CFD prediction results. The pressure drop across the bed predicted by ANN is subsequently lower than in CFD. This indicates that the solid inventory predicted by ANN is less than the actual amount of solid in the bed (as predicted by CFD results). This fact shows that the ANN can mimic the data used in the training, but that it has not learnt enough to be used as a predictive model. It means that the network does not have enough information to predict a blind test. To resolve this issue, an additional parameter—average pressure at the downstream or upstream layer—is added to the training data-set.

### 4.3. Training for Gas Pressure Using Static and Dynamic Parameters

To improve the quality of the trained ANN, one dynamic parameter (gas pressure) is added to the training data as the fifth input. During the training process, to train ANN for level "L,"the pressure from CFD results at level "L-1" (upstream level) is used as input. The training scheme discussed above is shown in Figure 36. The direction of ANN training was selected to be from fluidized bed inlet to outlet (upstream to downstream). During the training process, all data is actual CFD data; however when it comes to deployment, depending on the source of data (CFD or smart proxy), two different scenarios are available: non-cascading and cascading.

| 5 Inputs | | | | | | |
|---|---|---|---|---|---|---|
| Layer index | Dis. To top | Dis. To bottom | Inlet Vel. | Pressure From CFD at time step t | | |
| ✓ | ✓ | ✓ | 0.6 (m/s) | Averaged Gas Pressure at level"L-1" at $V_{in}$ = 0.6 (m/s) | | |
| ✓ | ✓ | ✓ | 0.75 (m/s) | Averaged Gas Pressure at level"L-1" at $V_{in}$ = 0.75 (m/s) | | |
| ✓ | ✓ | ✓ | 1.2 (m/s) | Averaged Gas Pressure at level"L-1" at $V_{in}$ = 1.2 (m/s) | | |

**Figure 36.** ANN deployment for average gas pressure using five parameters in cascading mode when information flows from upstream to downstream.

In this scenario, every layer is dependant on the previous layer; thus, values from the previous layer are required for the deployment process. In the non-cascading approach, results could be deployed very quickly because all of the data is from CFD and CFD data is available. However, in the cascading approach, ANN should be deployed for every layer separately because the output of one layer is the input of next layer. As discussed earlier, the cascading approach should be used for the deployment because the objective of Smart Proxy is to predict the gas pressure of the blind runs with any given inlet velocity.

The deployment sequence for the cascading approach is shown in Figure 37. Since the direction of deployment for layer "L" is from upstream to downstream, the bottom layer ($J = 1$) acts as the boundary condition and $P_{J=2}$ could be obtained using $P_{J=1}$. Then, $P_{J=3}$ is predicted using $P_{J=2}$ and this process will be repeated until it reaches to the top layer ($J = 162$).

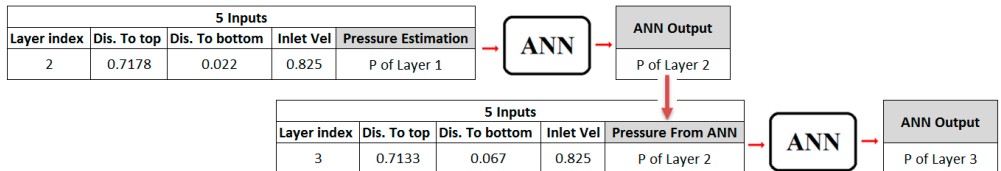

**Figure 37.** Detail of cascading deployment for average gas pressure using five parameters when information flows from upstream to downstream.

The only input that is required for cascading deployment besides the static parameters is the starting pressure of the bottom layer (inlet plane). Figure 38 shows the starting layer and deployment direction for predicting the average pressure at each layer for the cascading approach.

Since the pressure at the bottom layer is not known (not a prescribed boundary condition), the average value of pressure at Layer 1 from all CFD simulations is used as a reasonable starting point as shown in Figure 39. This average value is 102,644 Pa and will be the pressure estimate for Layer 1 during cascading deployment.

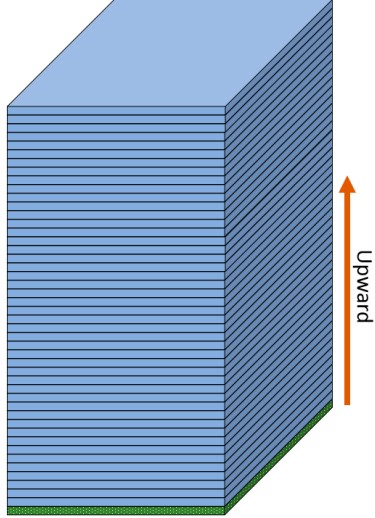

**Figure 38.** Starting layer (inlet plane) and direction of cascading deployment.

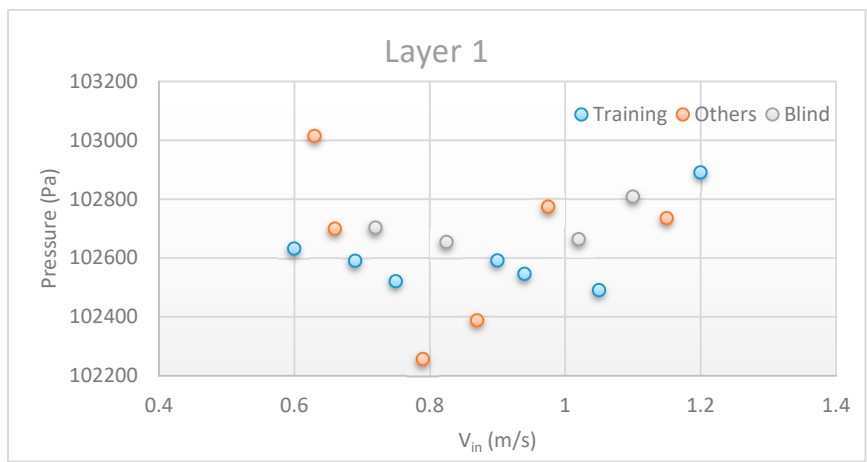

**Figure 39.** Average pressure across the bottom layer at different inlet velocities.

The smart proxy has been trained and deployed for the inlet velocity of 0.6 m/s that was a part of training in the cascading fashion. Figure 40 shows the agreement between the smart proxy and CFD.

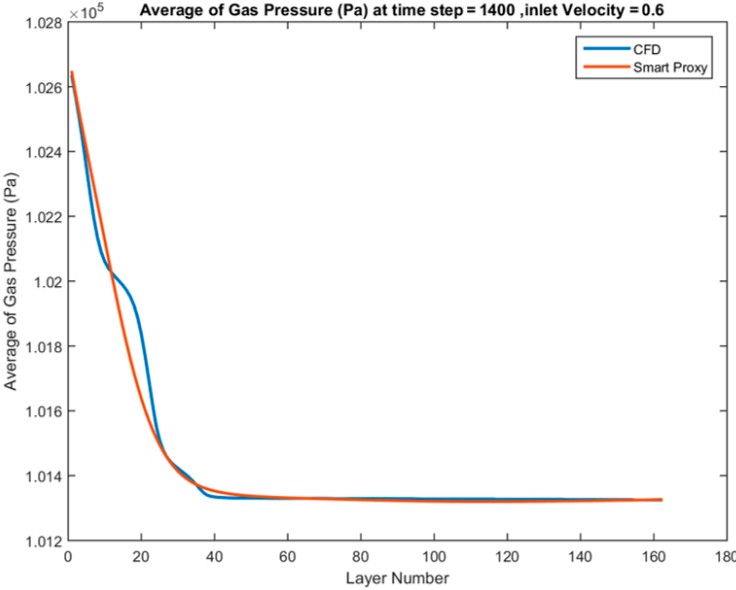

**Figure 40.** Spatially averaged CFD and smart proxy results for gas pressure (Pa) at time step = 1400 and $V_{in}$ = 0.6 m/s.

Next, the ANN is used for the blind test cases to evaluate the smart proxy predictability. Figure 41 shows the comparison between CFD results and ANN prediction in upstream to downstream cascading deployment when the $V_{in}$ = 0.825 m/s from blind test cases is used. As shown, the smart proxy and CFD have a good agreement. The only input to the ANN is the inlet velocity and the average gas pressure at the starting layer of $J$ = 1. The gas pressure at the starting layer is assumed to be 102,644 kPa, which is the average pressure at Layer 1 from all CFD simulations (Figure 39).

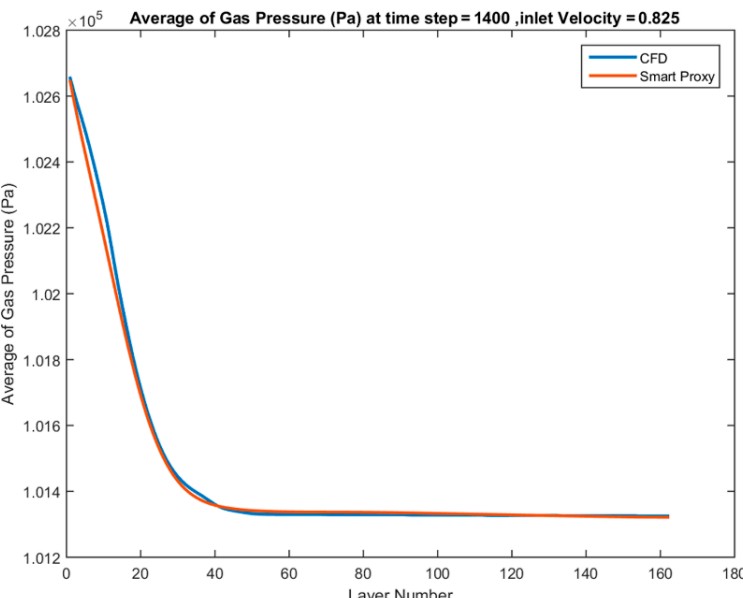

**Figure 41.** Spatially averaged CFD and smart proxy results for gas pressure (Pa) at time step = 1400 for blind test condition of $V_{in}$ = 0.825 m/s.

Figures 42 and 43 show the predictive capability of the trained ANN when deployed for two additional blind test cases with inlet velocities of $V_{in}$ = 0.72 and 1.02 m/s and time-step of 1400.

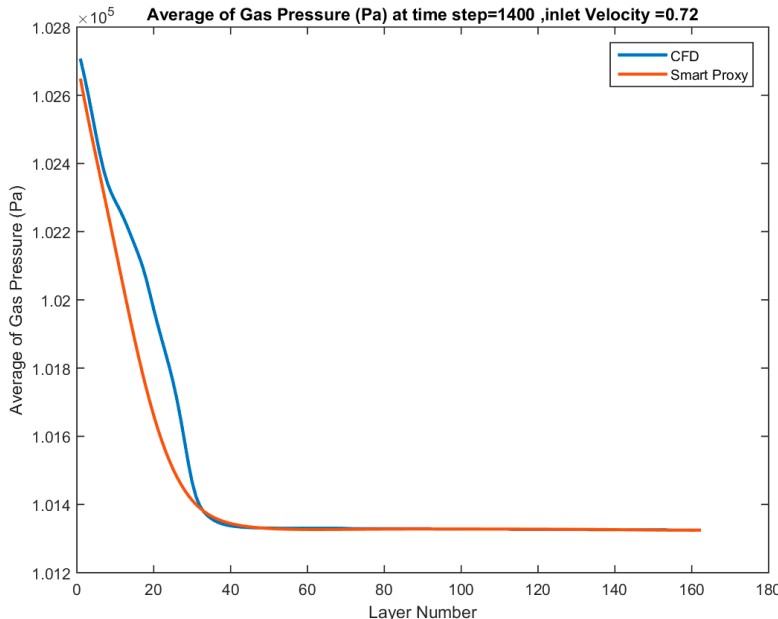

**Figure 42.** Spatially averaged CFD and smart proxy results for gas pressure (Pa) at time step = 1400 for blind test condition of $V_{in}$ = 0.72 m/s.

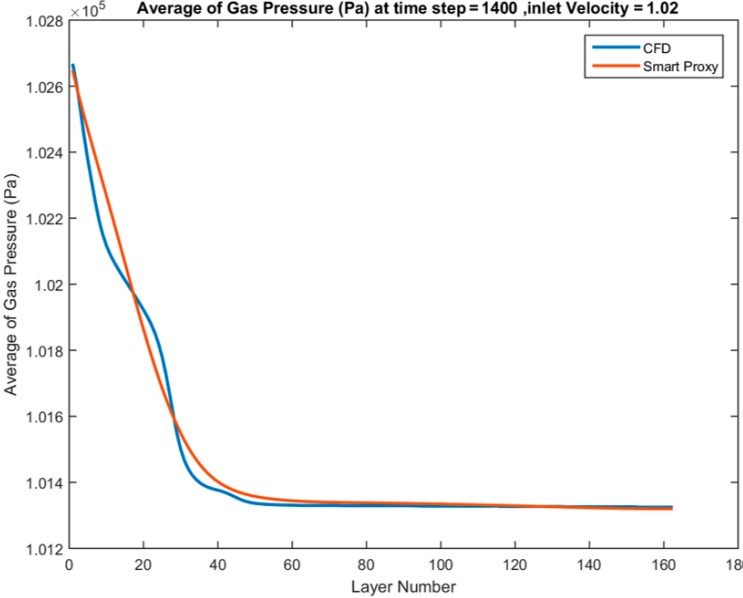

**Figure 43.** Spatially averaged CFD and smart proxy results for gas pressure (Pa) at time step = 1400 for blind test condition of $V_{in}$ = 1.02 m/s.

The above results show an acceptable agreement between CFD and smart proxy results. The smart proxy results show some deviation from CFD results in the bed. This may be due to the fact that pressure at the starting layer is the average of all CFD simulations and may differ from the actual average pressure at Layer 1 for individual cases.

### 4.4. Time average

Flow in a fluidized bed is highly transient and chaotic. As such, even multiple CFD simulations of the same flow conditions yield different instantaneous flow fields, although the time-averaged flow field must be the same. For this reason, the performance of a fluidized bed is typically assessed

based on the time and/or Spatially averaged behavior of the various variables such as pressure, velocities, and volume fraction of gas and solid particles.

More than one ANN is needed in order to perform time averaged analysis of the smart proxy results. This is achieved by constructing 10 ANNs for time steps 500 to 1400 at an increment of 100 time steps, and by constructing 20 ANNs for time steps 1500 to 3400 at an increment of 100 time steps using the training approach outlined in Figure 36. Each time step is 0.001 seconds of simulation time. Figure 44 shows the two-time periods used for time averaging, for time steps 500 to 1400, and for time steps 1500 to 3400, representing flow conditions depicted in Figures 9a and 9b and Figures 9c and 9d, respectively.

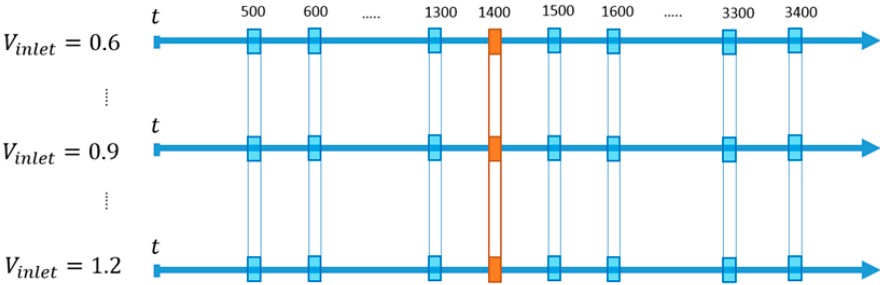

**Figure 44.** Time steps selected for time average.

Figure 45 shows all the smart proxy results for different time-steps for an inlet velocity of 0.825 m/s (a blind test case). All time-steps have the same average pressure of 102,644 Pa as their starting point at Layer 1. The black curve in Figure 45 shows the time averaged curve for time steps 500 to 1400.

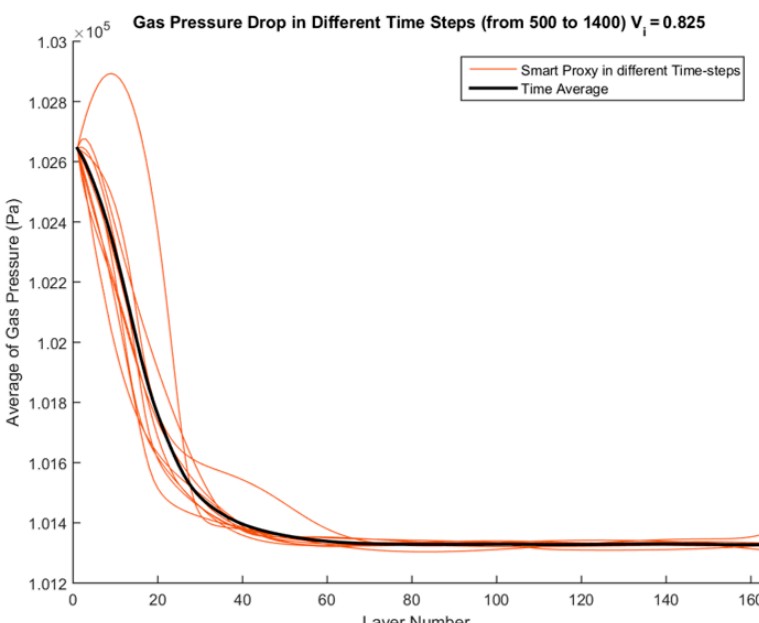

**Figure 45.** Spatial averaged profile of smart proxy results for gas pressure (Pa) for times 500 through 1400 and blind test condition of $V_{in}$ = 0.825 m/s.

Comparison between the time-averaged CFD results across each layer and time-averaged ANN results across each layer is shown in Figure 46 for $V_{in}$ = 0.825 m/s. The RMSE in Figure 46 is 38 Pa, which is less than 3% of the pressure drop across the fluidized bed. Additional deployments for blind test cases of inlet velocities of 1.02 and 1.1 m/s are shown in Figures 47 and 48, respectively.

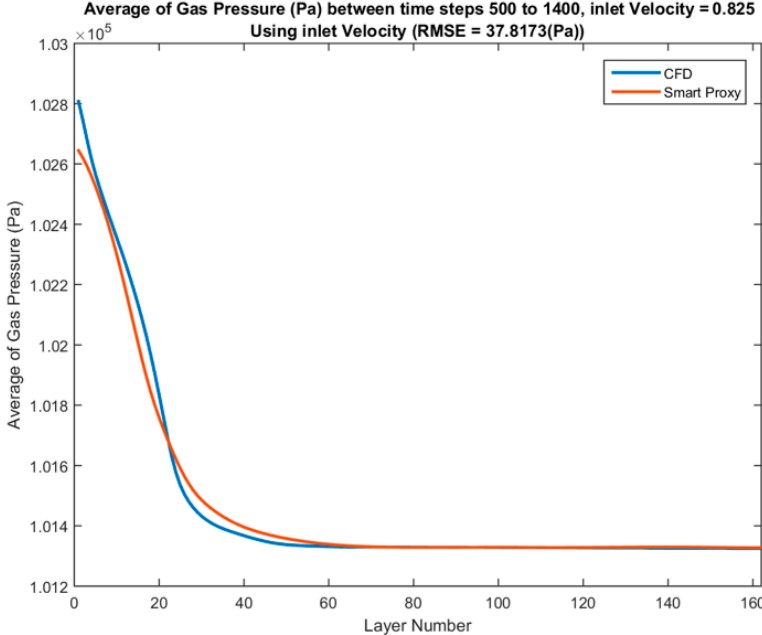

**Figure 46.** Spatial average profile of CFD and smart proxy results for gas pressure (Pa), averaged over time steps 500 to 1400 at blind test condition of $V_{in}$ = 0.825 m/s.

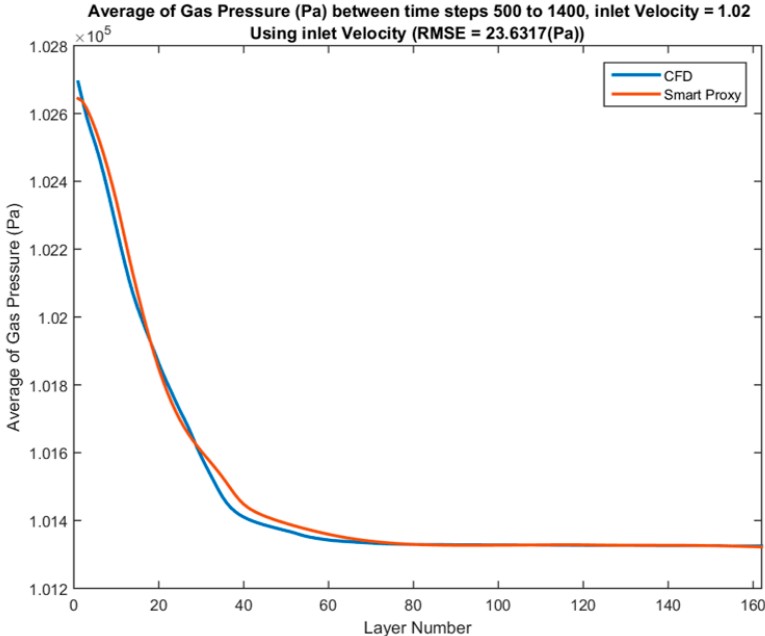

**Figure 47.** Spatial average profile of CFD and smart proxy results for gas pressure (Pa), averaged over time steps 500 to 1400 for blind test condition of $V_{in}$ = 1.02 m/s.

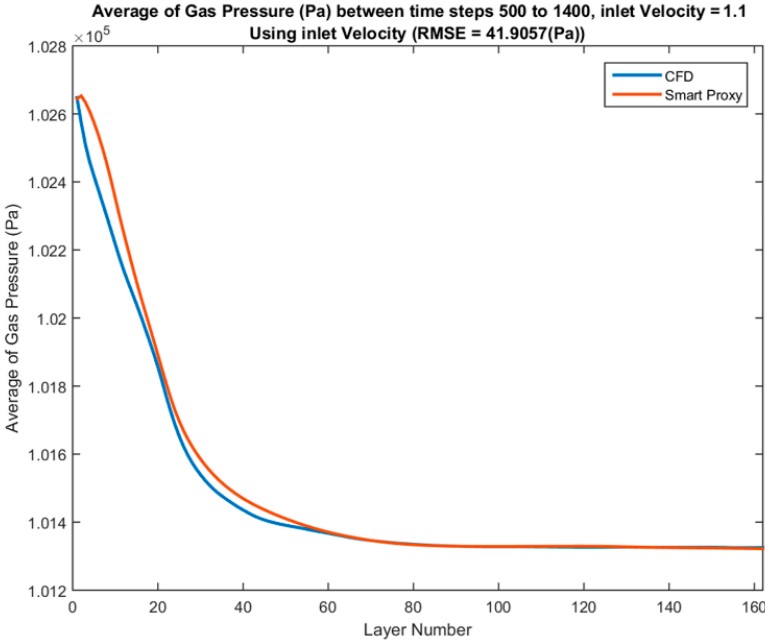

**Figure 48.** Spatial average profile of CFD and smart proxy results for gas pressure (Pa) averaged over time-steps 500 to 1400 for blind test condition of $V_{in}$ = 1.1 m/s.

The smart proxy model is able to capture the behavior of time-steps 500 through 1400 reasonably well when flow is more of a slugging flow. A series of ANNs are constructed at time-steps 1500 to 3400 at an increment of 100 to illustrate the performance of the ANN when flow becomes fluidized. Blind tests are carried out once the ANNs are trained.

Figure 49 shows all the smart proxy results for different time-steps for inlet velocity of 0.825 m/s (a blind test case). The black curve in Figure 49 shows the time-averaged pressure across all layers between time steps 1500 to 3400.

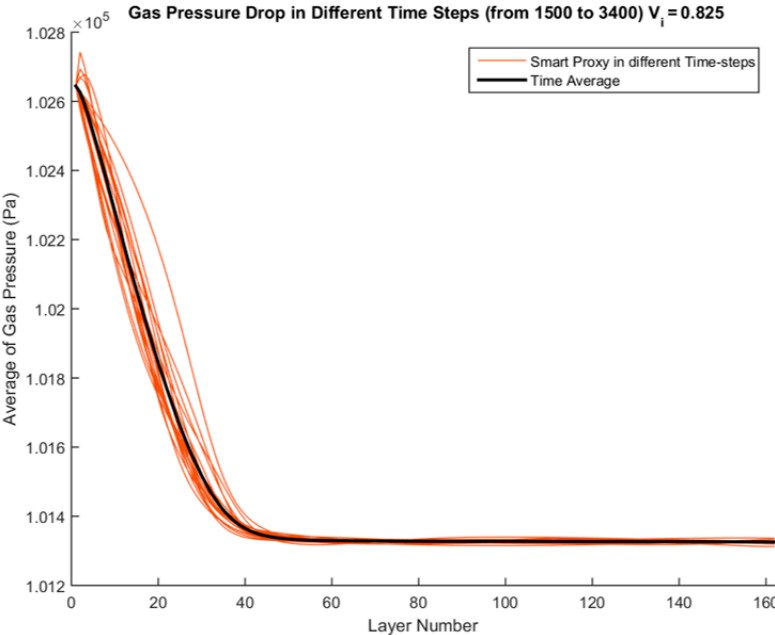

**Figure 49.** Spatial average profile of smart proxy results for gas pressure (Pa) for time steps 1500 to 3400 at blind test condition of $V_{in}$ = 0.825 m/s.

The comparison of time-averaged pressure values across each layer from CFD results and ANN prediction in deployment mode for three blind test cases with inlet velocities of 0.825, 1.02 and 1.1 m/s between time steps of 1500 to 3400 are shown in Figures 50–52, respectively.

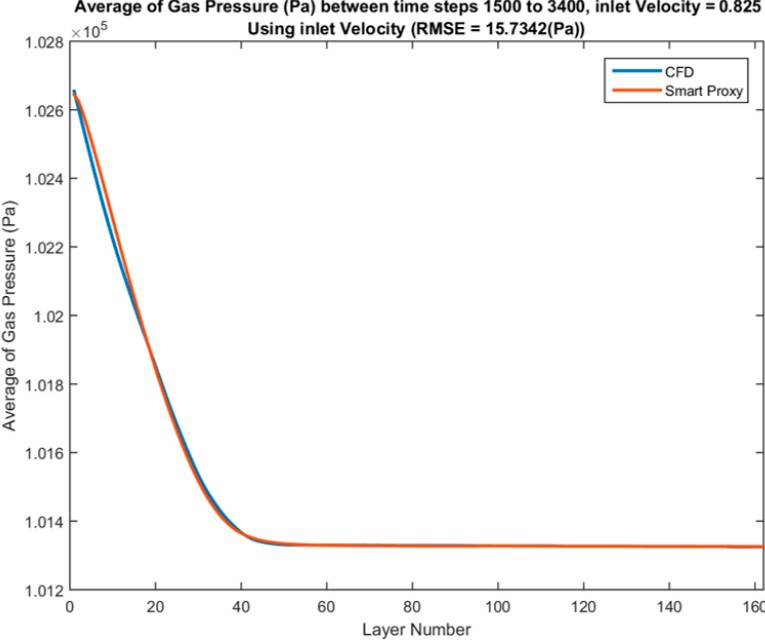

**Figure 50.** Spatial average profile of CFD and smart proxy results for gas pressure (Pa) averaged over time-steps 1500 to 3400 for blind test condition of $V_{in}$ = 0.825 m/s.

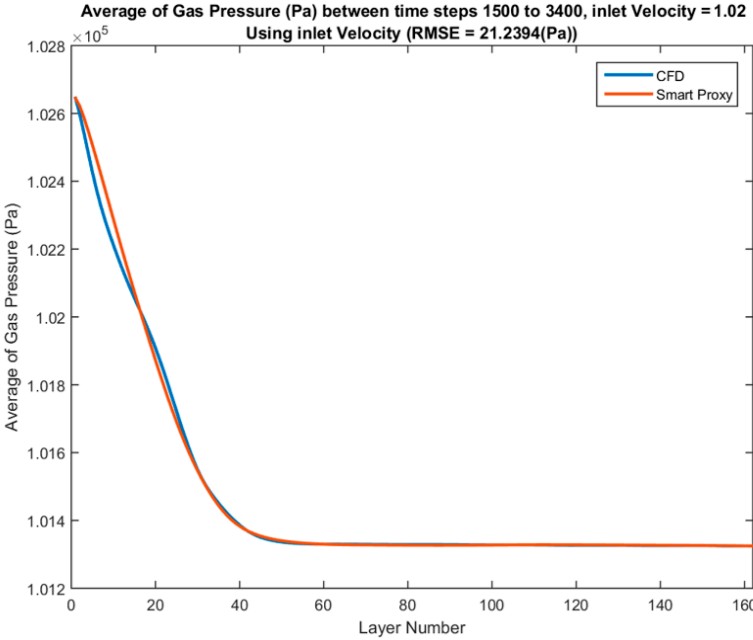

**Figure 51.** Spatial average profile of CFD and smart proxy results for gas pressure (Pa) averaged over time-steps 1500 to 3400 for blind test condition of $V_{in}$ = of 1.02 m/s.

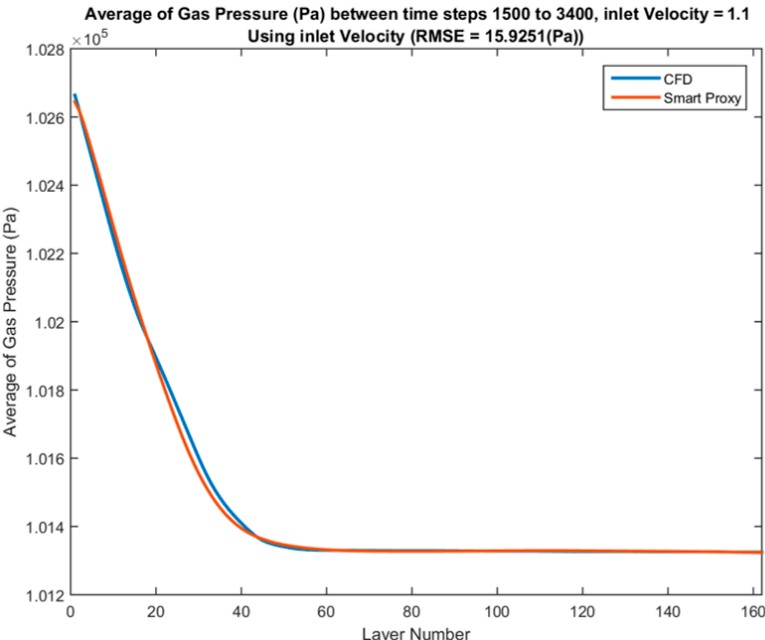

**Figure 52.** Spatial average profile of CFD and smart proxy results for gas pressure (Pa) averaged over time steps 1500 to 3400 for blind test condition of $V_{in}$ = 1.1 m/s.

## 5. Conclusion

A data-driven smart proxy was developed to mimic the CFD simulation results of a three-dimensional fluidized bed with good accuracy and faster speed. Table 5 shows the comparison of data preparation and run time of these two approaches. On average, training of ANN takes between 24 hours to 36 hours depending on the scenario under consideration. The training time of an ANN is also strongly affected by the computer hardware. The developed model can be used to generate the time-averaged results of any given fluidized bed with the same geometry with different inlet velocity in 180 s. This is considerably shorter compared to CFD execution time (a wall time speedup of 1440%). This study shows that machine learning and artificial intelligence can be important tools in multiphase flow modeling and warrant further investigation.

**Table 5.** Comparison between speed of run for CFD and smart proxy.

| Method | Task | Required Time |
|:---:|:---:|:---:|
| CFD | Modeling and Simulation Time | 3 days on 4 CPUs |
| | Data Preparation (CFD simulation) | 3 days for each case |
| Smart Proxy | Model Training | 24 to 36 hours |
| | Generating the results for a new case | 180 s on 1 CPU |

The first part of this study showed that the smart proxy could be a viable tool for predicting gas-solid flow behavior in a fluidized bed. Additionally, to make the trained ANN more general, the cascading deployment needs further research in order to minimize the error propagation over time.

In the second part, a different approach was utilized to develop another data-driven smart proxy at the layer level by treating the time-step in a different fashion. The average time to train one ANN was about 1 minute. This model had a reasonable accuracy and faster execution time.

This research has shown that ANN can be a powerful tool for analyzing gas-solids behavior of a fluidized bed within the operating condition range used during training of ANN. As such, the operating range of ANN is limited to conditions that are used during training. ANN cannot be used to extrapolate beyond the training operating envelope.

**Author Contributions:** Conceptualization, S.M., and M.S; Methodology, A.A. and S.M.; Software, M.S., and A.A.; Validation, A.A.; Data Curation, M.S, and A.A.; Writing-Original Draft Preparation, A.A.; Writing-Review & Editing, A.A., M.S, and J.D.; Visualization, A.A.; Supervision, S.M., M.S., and J.D.; Project Administration, S.M., and M.S.; Funding Acquisition, S.M., and M.S.

**Funding:** S. D. Mohaghegh acknowledges the support provided by an appointment to the National Energy Technology Laboratory (NETL) Faculty Research Participation Program, sponsored by the U.S. Department of Energy and administered by the Oak Ridge Institute for Science and Education. This work is performed as part of NETL research for the U.S. Department of Energy's Cross Cutting Program.

**Acknowledgments:** The authors acknowledge computational resources provided by NETL High Performance Computing.

**Conflicts of Interest:** The authors declare no conflict of interest.

## Appendix A

This appendix describes the fluid-solid and solid-solid momentum transfer models between the phases used in MFiX.

*Fluid-Solids Momentum Transfer*

The fluid-solids interaction force is a combination of buoyancy, the drag force, and momentum transfer due to mass transfer.

$$\vec{I}_{gm} = -\epsilon_{sm}\nabla P_g - F_{gm}\left(\vec{v}_{sm} - \vec{v}_g\right) + R_{0m}(\xi_{0m}\vec{v}_{sm} - \vec{\xi}_{0m}\vec{v}_g) \tag{A1}$$

$R_{0m}$ is the mass transfer from the gas phase to the mth solid phase, where

$$\xi_{0m} = \begin{cases} 1 \; for \; R_{0m} < 0 \\ 0 \; for \; R_{0m} \geq 0 \end{cases} \tag{A2}$$

$$\vec{\xi}_{0m} = 1 - \xi_{0m}$$

$F_{gm}$ is the coefficient for the interphase force between the fluid phase and the mth solid phase.

$$F_{gm} = \frac{3\epsilon_{sm}\,\epsilon_g\,\rho_g}{4\,V_{rm}^2\,d_{pm}}\,C_{DS}(\frac{Re_m}{V_{rm}})\left|\vec{v}_{sm} - \vec{v}_g\right| \tag{A3}$$

Re$_m$ is Reynolds number of mth phase:

$$Re_m = \frac{d_{pm}\left|\vec{v}_{sm} - \vec{v}_g\right|\rho_g}{\mu_g} \tag{A4}$$

$V_{rm}$ is the terminal velocity correlation for the mth solids phase which is only a function of gas volume fraction.

$$V_{rm} = 0.5\left(A - 0.06\,Re_m + \sqrt{(0.06\,Re_m)^2 + 0.12Re_m(2B - A) + A^2}\right) \tag{A5}$$

Where the coefficient A and B are calculated as follows.

$$A = \epsilon_g^{4.14} \tag{A6}$$

$$B = \begin{cases} 0.8 \, \epsilon_g^{1.28} \; if \; \epsilon_g \leq \; 0.85 \\[2mm] \epsilon_g^{2.65} \; if \; R_{0m} \; > 0.85 \end{cases}$$

*Solids-Solids Momentum Transfer*

The solids-solids momentum transfer comes from the drag force between different phases and is calculated using the equation below.

$$\vec{I}_{ml} = -F_{sml}(\vec{v}_{sl} - \vec{v}_{sm}) + R_{ml}(\xi_{ml}\vec{v}_{sl} + \vec{\xi}_{ml}\vec{v}_{sm}) \tag{A7}$$

$R_{ml}$ is the mass transfer from mth solid phase to 1st solid phase.

$$\xi_{ml} = \begin{cases} 1 \, for \, R_{ml} < \; 0 \\[2mm] 0 \, for \, R_{ml} \; \geq 0 \end{cases} \tag{A8}$$

$$\vec{\xi}_{ml} = 1 - \; \xi_{ml}$$

The drag coefficient $\vec{F}_{sml}$ is defined as follows.

$$\vec{F}_{sml} = \frac{3(1 + e_{lm})\left(\frac{\pi}{2} + \frac{C_{flm}\pi^2}{8}\right)\epsilon_{sl}\,\rho_{sl}\,\epsilon_{sm}\,\rho_{sm}\,(d_{pl} + d_{pm})^2 \; g_{0lm}\,|\vec{v}_{sl} - \vec{v}_{sm}|}{2\pi\left(\rho_{sl}\,d_{pl}^3 + \; \rho_{sm}\,d_{pm}^3\right)} \tag{A9}$$

Where e<sub>lm</sub> and $C_{flm}$ are the coefficient of restitution and coefficient of friction, respectively, between the lth and mth solids-phase particles. $g_{0lm}$ is the radial distribution function at contact.

$$g_{0lm} = \frac{1}{\epsilon_g} + \frac{3\left(\sum_{\lambda=1}^{M}\frac{\epsilon_{s\lambda}}{d_{p\lambda}}\right)d_{pl}d_{pm}}{\epsilon_g^2\,(d_{pl} + d_{pm})} \tag{A10}$$

More details and explanations can be found in the MFiX documentation [25,26].

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
