# Peer review of "Modeling Average Pressure and Volume Fraction of a Fluidized Bed Using Data-Driven Smart Proxy"

_fluids, doi:10.3390/fluids4030123_

Round 1
Reviewer 1 Report
In this paper, the authors developed a strategy for predicting solutions of CFD models, in particular, for gas-solid multiphase flows, based on data-driven techniques, namely Artificial Neural Nets. I believe this paper brings up an important contribution to the field, as recently, a tremendous growth in assessing the feasibility of machine learning techniques to enhance and mitigate the cost of performing high fidelity simulation has been observed in all areas of engineering.
I believe this paper has good a potential to warrant publication, but major revision is needed to clarify and enhance many of the results presented here. See below my assessment:
1) To begin with the authors need to make sure the abstract reflects the main findings of the papers. It has been written in very general terms (it sounds more like a note in a report to sponsors), even mentioning NETL participation that there is not a directly connection with the scientific findings presented here. They should state briefly the contribution, methodology and results.
2) In general, the scientific findings seems to be solid and in line to what one should expect from ANN, but I believe the authors should expand on their literature and citation exposition. Most (if not all) of the citation regarding ANN to fluids have been referenced to the same research group, which does not reflect the contributions in this area in the last 10-15 years. Authors should demonstrate that this contribution is indeed part of a bigger effort in mitigating computational cost and they are not the only one working on this.
3) As a follow up on item 2 above, statements like in page 2/line 90, namely, "... The idea of using AI in fluid mechanics was first introduced by ..." is not simply true! Proxy modeling/AI has been a very active area in CFD, especially in the aerospace industry for a long time. I can cite a an old book (see below), but many publications in this area date back from the 90's. The authors should modify this statement and make sure proper references are cited.
Computational Approaches for Aerospace Design: The Pursuit if Excellence. 2005
4) Regarding their mathematical modeling, section 2.1 is a little bit shy of providing a good understanding the physics of the problem. The equations are not explained, and I not really sure what are the inputs/outputs of the problem,. The authors mention this in section 2.4, it seems later to make a connection to the physical problem. Also, the authors do not describe in details the discretization of such PDE's as this may be important to make connections to Fig. 3. I would suggest a simple appendix with more information about the MFiX setup.
5) Section 2.2 seems disconnected from the idea of the paper. So what ? What are you trying to introduce with the section in ML?
6) In section 2.4 the authors mention the order of the input/output is important for good training. Can you expand on this statement and show why? It seems that you are already creating/inputing a connection between cells using the location parameters, so your ANN should be able to match this information. Also, it is not clear the purpose of your tier system. It seems you are trying to find connections between cells, but again, your discretization approach would play a role on this, and this has not been investigated further. Please clarify.
7) It is not clear to me why only one time step at a time is being used as input and output. Well, after running your high fidelity model - from my understanding, you should run it at least once, but needs multiple runs for multiple scenarios - you have already all time steps in the model. If this is not correct, please clarify.
In any case, if you have your time series, why this cannot be your input ? Is your ANN not able to retrieve the evolution of the dynamical system? In section 3, you mention that you are outputting one step at a time and one property at a time? Why? Is your ANN not able to predict the parameters of table 1 simultaneously?
8) Predicting the states of Fig. 11,12, 15-18, requires a deep knowledge of the exact time steps one should train the model. This means that a rigorous analysis of the high fidelity model needs to be performed before hand. Why have you selected the time steps in these figures? The same holds for the the cases presented in section 3.3. This requires understanding of the regime changes which are not know a priori. Can the authors clarify and expand on this issue ?
9) From my understanding, the authors fixed the ANN structure using the same characteristics as in Table 2 and using the same number of hidden layers/neurons. It is not clear if the authors assessed the sensitivity to these parameters. Please comment.
10) In the the cascading mode, errors will accumulate, and in the limit they may grow unbounded. Have you experienced anything like that? If not, can you explain what limits the error. If you did, can you provide remediations?
Also, Figure 14 is really hard to see, and the text right below (not figure caption) seems in a different font.
11) Regarding Table 4 - the times shown are not representative of the actual simulation times. Although the authors report ANN training in the order of 24-36 hours, it is not clear if this includes running the fine scale/high fidelity simulator multiple times (multiple scenarios) to gather the input to ANN. I suggest the authors present a breakdown of time/effort for every step of the process.
Due to all of my concerns/suggestions and the fact this is a good contribution to the area, I recommend this paper be published after major revisions.
Author Response
Dear Reviewer,
Thanks for taking your time and going through the first draft of our manuscript. Your comments adds a lot of value to this article. I tried to answer all of your comments and make modification to the manuscript accordingly. I have updated the abstract, literature review, MFiX equations and discretization method. I will attach my responses to you in a separate word file, I hope you find my responses helpful.
Regards

Reviewer 2 Report
Please see attached

Author Response
Dear Reviewer,
Thanks for taking your time and going through the first draft of our manuscript. Your comments adds a lot of value to this article. I tried to answer all of your comments and make modification to the manuscript accordingly. I have updated literature review and added the color bar to two figures as you suggested. I attached my responses to you in a separate word file, I hope you find my responses helpful.
Regards

Reviewer 3 Report
This paper presents a machine learning based method to train a data-driven smart proxy for predicting the pressure and volume fraction in a fluidized bed. It is an interesting work and the smart proxy demonstrate its value by reducing the computing time significantly compared to traditional CFD methods. The proposed method also achieves reasonable accuracy with error less than 10%. Generally, I found this manuscript is well written and detailed. However, I have a few questions that I would like the author to explain in more details.
1. My main concern lies in the design of the input parameters for training the ANN at cell level. In 2.4.1, the author states that the different tiers of adjacent cells can be chose for training the ANN dependent on the complexity of the problem. What is the justification in this study using 6 cells rather than using 18 or 26 cells?
2. It is good to consider the possible influence of cell locations in training the neural network. However, in the training process, even in multiple time-steps, three are only three unique pair of dynamic parameter vectors (1x63) for each specific location. If these dynamics parameters can have large variation at each spatial location, with only three training points in each location, it is not possible for ANN to learn these corresponding. Anther possibility is that the output of the ANN is actually not sensitive to the location. It can mainly dependent on the 63 dynamics parameters.
The reviewer suggest the author to add additional test for verification. One way is to fix the 63 parameters and vary cell locations parameter (six) to check the output response. This can verify does the location really have any influence on output in current trained ANN. If the output is location dependent, the current training data may not be sufficient to train a reliable ANN, since for each location, there are only three at most data points. If it is not heavily dependent on location, one can consider removing location as the input.
3. The notation of the output in Figure 6, Figure 10 and Figure 19 can be misleading. In the training process, the ANN can generate an output with given input. This output should be compared with the CFD results with the same input value. With this calculated value, the ANN can update its parameters using the back propagation. The figures should be modified for clarity.
4. It is unclear how the layer K is defined as shown in Figure 11 and Figure 12.
Author Response

(The authors gave the same response as above.)

Round 2
Reviewer 1 Report
I would like to thank the authors for working on my concerns and suggestions. The paper reads better than the previous version.
I would just suggest for the authors to be more clear (perhaps color the text or add comment in the response to the reviewer) with respect to their changes in the revised manuscript. It was hard to find the new/edited phrases, paragraphs and explanations.
Author Response
Dear Reviewer,
Thanks for taking your time and going through our manuscript. Your comments adds a lot of value to this article.
Regards
Reviewer 2 Report
The manuscript is accepted.
Author Response

(The authors gave the same response as above.)

Reviewer 3 Report
The authors cleared my questions. I would like to recommend this paper for publication.
Author Response

(The authors gave the same response as above.)
